# A genetically encoded BRET-based SARS-CoV-2 M^pro protease activity sensor

Anupriya M. Geethakumari [1], Wesam S. Ahmed [1], Saad Rasool[2], Asma Fatima[1], S. M. Nasir Uddin[1], Mustapha Aouida [1] & Kabir H. Biswas [1✉]

The main protease, M^pro, is critical for SARS-CoV-2 replication and an appealing target for designing anti-SARS-CoV-2 agents. Therefore, there is a demand for the development of improved sensors to monitor its activity. Here, we report a pair of genetically encoded, bioluminescence resonance energy transfer (BRET)-based sensors for detecting M^pro proteolytic activity in live cells as well as in vitro. The sensors were generated by sandwiching peptides containing the M^pro N-terminal autocleavage sites, either AVLQSGFR (short) or KTSAVLQSGFRKME (long), in between the mNeonGreen and NanoLuc proteins. Co-expression of the sensors with M^pro in live cells resulted in their cleavage while mutation of the critical C145 residue (C145A) in M^pro completely abrogated their cleavage. Additionally, the sensors recapitulated the inhibition of M^pro by the well-characterized pharmacological agent GC376. Further, in vitro assays with the BRET-based M^pro sensors revealed a molecular crowding-mediated increase in the rate of M^pro activity and a decrease in the inhibitory potential of GC376. The sensors developed here will find direct utility in studies related to drug discovery targeting the SARS-CoV-2 M^pro and functional genomics application to determine the effect of sequence variation in M^pro.

[1] Division of Biological and Biomedical Sciences, College of Health & Life Sciences, Hamad Bin Khalifa University, Education City, Qatar Foundation, Doha 34110, Qatar. [2] Division of Genomics and Precision Medicine, College of Health & Life Sciences, Hamad Bin Khalifa University, Education City, Qatar Foundation, Doha 34110, Qatar. ✉email: kbiswas@hbku.edu.qa

COVID-19 has become a global health threat with more than 300 million infections and more than 5 million deaths as of January 2022. The causative agent, Severe Acute Respiratory Syndrome Corona Virus 2 (SARS-CoV-2) of the *beta-coronavirus* family shares 79% similarity with SARS-CoV and 50% similarity with MERS-CoV (Middle East respiratory syndrome coronavirus)[1,2]. SARS-CoV-2 infection cycle is initiated by the processing of two polypeptides, pp1a and pp1ab, bearing the non-structural proteins[3] by the auto-catalytically released viral proteases, 3-chymotrypsin-like cysteine protease (3CL^pro) or main protease (M^pro)[4,5], and papain-like protease (PL^pro)[6–8]. M^pro functions as a homodimer with each monomer containing an active site formed by a conserved catalytic dyad of Cys-His[7,9], and cleaves the large polyprotein pp1ab at 11 sites[7]. Specifically, M^pro recognizes a highly conserved core sequence with a critical Gln residue for cleavage[10–13]. Importantly, M^pro cleavage sequences are not known to be recognized by human proteases, thus making M^pro an attractive target for anti-SARS-CoV-2 therapy[14].

Given the critical role played by M^pro in SARS-CoV-2 infection and the cleavage specificity, a number of assays have been developed to monitor the proteolytic activity of M^pro. Genetically-encoded reporters based on either fluorescence or bioluminescence provide sensitive and specific means to assess biologically relevant events such as cell signaling, conformational changes of proteins and protein-protein interactions in live cells[15–18]. Researchers have developed fluorescence and bioluminescence-based reporter assays for screening antiviral molecules against various coronaviruses (e.g., fluorescence resonance energy transfer (FRET)-[19] and split-luciferase-[20,21] based assays). Specifically, a number of studies have utilized FRET-based in vitro assays, wherein peptide substrates containing the M^pro cleavage sequences are used as reporter, for the identification of antivirals against SARS-CoV-2 M^pro[22–26]. Additionally, a FRET-based assay was utilized for the identification of Boceprevir, GC376, and calpain inhibitors II, XII as potent inhibitors of SARS-CoV-2 M^pro[27]. On the other hand, a FlipGFP-based construct containing the M^pro N-terminal autocleavage site has been developed to screen the antivirals against SARS-CoV-2[28,29]. In such a construct, M^pro-mediated cleavage of FlipGFP in live cells results in the generation of the fluorescent form of GFP from a non-fluorescent form. In a later report, authors have combined FlipGFP and luciferase assays to successfully identify additional M^pro inhibitors[29].

In addition to the above, Bioluminescence Resonance Energy Transfer (BRET) has been used in developing a range of genetically encoded, live cell sensors[30–33]. BRET relies on the non-radiative resonance energy transfer from a light emitting luciferase protein (donor) upon oxidation of its substrate to a fluorescent protein (acceptor) with an excitation spectrum overlapping with the luciferase emission spectra. In addition to the spectral overlap, BRET also depends on the physical distance and relative orientation of the donor and the acceptor proteins[17,18,34,35]. The latter has been successfully utilized in generating a variety of molecular sensors including detecting small molecules[18,36–38], structural changes in proteins[35,37]. While a number of donor-acceptor pairs with distinct spectral and energy transfer efficiencies have been utilized for BRET-based sensor development[39], the combination of mNeonGreen (mNG)[40–43], a bright green fluorescent protein, and NanoLuc (NLuc)[44,45], a small bright and stable luciferase have gained significant usage in the recent times including proteolytic cleavage sensors[38,46] due to excellent spectral overlap and light emission characteristics[47–53].

In the present study, we have engineered BRET-based M^pro proteolytic activity sensors by inserting the M^pro N-terminal autocleavage sequences (either the short AVLQSGFR[4,26] or the long KTSAVLQSGFRKME[5,26] in between the mNG (acceptor) and NLuc (donor) in a single fusion construct. The sensor constructs showed robust cleavage activity in live cells when coexpressed with the wild type M^pro but not in the presence of the catalytically dead C145A mutant M^pro[54–56], and with a faster kinetics and higher specificity compared to the recently reported FlipGFP-based M^pro sensor. We have further determined the utility of the sensors in pharmacological inhibition of the M^pro using the well-established M^pro inhibitor, GC376[27,57–59]. Additionally, in vitro assays showed a molecular crowding-mediated proteolytic cleavage rate enhancement and inhibitor potency decrease.

## Results and discussion

**BRET-based M^pro proteolytic cleavage activity sensor design.** In order to develop a live cell, BRET-based specific reporter to monitor M^pro proteolytic cleavage activity, we generated fusion proteins containing the M^pro N-terminal autocleavage sequence sandwiched between mNG (acceptor) and NLuc (donor) proteins (Fig. 1). The mNG and NLuc pair (acceptor and donor, respectively) has been used in a number of BRET-based sensors and show efficient energy transfer from NLuc to mNG. Thus, in the absence of any proteolytic cleavage, the sensor constructs are expected to display significant emission in the green channel. However, upon proteolytic cleavage of the sandwiched autocleavage peptide, the sensor constructs will display reduced emission in the green channel with a concomitant increase in the emission in the blue channel (Fig. 1).

Both SARS-CoV-2[10] and SARS-CoV-1[60] M^pro show a significant preference for the N-terminal autocleavage sequence (AVLQSGFR; short sensor; Figs. 1 and 2A; see Supplementary Methods for complete sequence of the sensor constructs) as a substrate compared to other cleavage sequences in the pp1a polyprotein in terms of catalytic efficiency, and has been widely utilized in FRET-based, in vitro assays[22,26,28,61] as well as in a FlipGFP-based, live cell assay[28,29]. Additionally, we analyzed all available pp1a polyprotein sequences reported for SARS-CoV-2 isolates at the NCBI Virus database for any variation in the cleavage sequence. This indicated that the N-terminal autocleavage sequence is invariable in all isolates reported and therefore, the sensor construct designed here will serve as a reliable reporter for M^pro proteolytic cleavage activity.

We note that while BRET comes with several advantages including a higher signal-to-noise ratio and an extended dynamic range compared to some other methods[62], the presence of the acceptor and donor proteins i.e., mNG and NLuc at the N- and C-termini, respectively could potentially affect the interaction of the cleavage peptide with the M^pro dimer and thus, in turn, affect the cleavage efficiency of the peptide. This is especially relevant given that the binding of the peptide substrate has been reported to allosterically activate the SARS-CoV-1 M^pro dimer[63,64]. Therefore, we generated a second, extended M^pro sensor construct the KTSAVLQSGFRKME peptide sequence (containing additional three residues on each sides of the AVLQSGFR core sequence; long sensor; Fig. 3B; see Supplementary Methods for complete sequence of the sensor construct), which has also been used in several previous reports[5,22,26]. A key requirement for efficient cleavage of peptide substrates by M^pro is their structural flexibility as previous studies have reported that the formation of secondary structural elements can alter the rate of their cleavage[65–67], especially given that secondary structure prediction indicated α-helical propensity by both the short as well as the long peptide (Supplementary Fig. 1). In order to assess structural flexibility and secondary structure formation by the two peptides, we generated structural models of the peptides using the

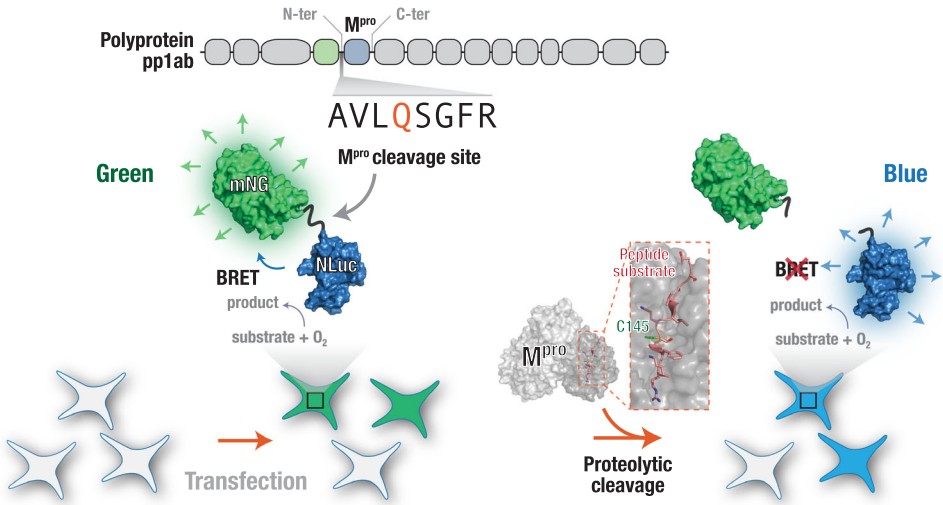

**Fig. 1 Genetically encoded, BRET-based live cell SARS-CoV-2 M$^{pro}$ protease activity sensor.** A schematic representation of the genetically encoded, BRET-based SARS-CoV-2 M$^{pro}$ protease activity sensor expressed in live cells. Close positioning of the NLuc (BRET donor) and mNG (BRET acceptor) proteins result in a significant resonance energy transfer in the absence of the SARS-CoV-2 M$^{pro}$ protease activity. Activity of the SARS-CoV-2 M$^{pro}$ results in the cleavage of the sensor resulting in a decrease in the resonance energy transfer between NLuc and mNG leading to a decrease in the green fluorescence of the sensor. Shown in gray is the surface representation of M$^{pro}$ structure (SARS-CoV; PDB: 2Q6G[68]) highlighting the critical C145 residue (green) required for proteolytic activity and the substrate peptide (TSAVLQSGFRK; red).

substrate peptide co-crystallized with the H41A mutant SARS-CoV-1 M$^{pro}$[68] and performed all-atom, explicit solvent, Gaussian accelerated molecular dynamics (GaMD) simulations. This allows enhanced sampling of protein conformational states by applying harmonic boost potential that follows a Gaussian distribution to accelerate conformational transitions[69,70]. Structural models were generated using Modeller[71] (Fig. 2A, B) and MD simulations were performed using the NAMD software[72] for a total duration of 1 μs for each peptide (Supplementary Movies 1 and 2). These simulations indicated significant structural fluctuations in the two peptides as revealed by relatively large root-mean-squared-deviations (RMSD) and root-mean-squared-fluctuations (RMSF) (Fig. 2C, D). Further, radius of gyration ($R_g$) measurements of the peptides over the course of these simulations also revealed structural fluctuations of the peptides with an appreciably greater fluctuations observed for the shorter peptide compared to the longer one (Fig. 2G, H). Additionally, this analysis also revealed a greater $R_g$ for the longer peptide compared to the shorter peptide (median $R_g$ values of 9.4 vs. 10.2 Å) as expected. Finally, secondary structure analysis of the peptides over the course of 1 μs long simulations revealed that the peptides have a propensity to form turns (Fig. 2I, J). Notably, certain central residues in the shorter peptide may form α-helix that was not seen with the longer peptide raising the possibility of a differential cleavage efficiency of the peptides by M$^{pro}$. These results were in agreement with dynamic light scattering (DLS) measurements that showed an average size of 6.2 ± 1.2 nm for the short M$^{pro}$ biosensor (Supplementary Fig. 1C), suggesting a compact structure of the sensor construct, which may show high BRET signal that is critical for detecting sensor cleavage. In the following, we report experimental results obtained with both the sensor constructs in order to provide a comparative analysis and determine the one that serves as a better substrate and thus, provide a better evaluation of M$^{pro}$ proteolytic cleavage activity.

**BRET-based M$^{pro}$ proteolytic cleavage activity sensor characterization in live cells.** In order to test the functionality of the BRET-based M$^{pro}$ proteolytic cleavage activity sensors, we

transfected HEK 293T cells with the short and long sensors either alone or along with the M$^{pro}$ expressing plasmid in a 1:5 sensor-to-protease plasmid ratio (Fig. 3). Additionally, we utilized the catalytically dead C145A mutant M$^{pro}$ as a negative control in these experiments since Cys145 is essential for the proteolytic activity of M$^{pro}$[54–56]. The transfection efficiency and expression of the sensor constructs was monitored by imaging live cells for mNG fluorescence using an epifluorescence microscope, which showed an efficient transfection and expression of the sensor constructs after 24 h of transfection (Supplementary Fig. 2).

We then determined the spectral properties of the two sensors in live cells. For this, sensor construct transfected cells in adherent conditions were incubated with NLuc substrate and emission in the range of 380 and 664 nm wavelength were detected using a microplate reader. In the absence of coexpression of M$^{pro}$, both the short and the long sensor showed two emission peaks corresponding to NLuc (467 nm) and mNG (533 nm), respectively, as determined from two Gaussian fitting of the spectral data (Fig. 3C, D; top panels). Coexpression of the wild type M$^{pro}$ resulted in a decrease in the mNG emission peak in cells expressing either of the sensor constructs (Fig. 3C, D; middle panel) while no such decrease was observed when the C145A mutant M$^{pro}$ was coexpressed with the sensor constructs (Fig. 3C, D; bottom panel). We note that the coexpression of either the wild type or the C145A mutant M$^{pro}$[54–56] did not result in any significant change in the intracellular levels of the sensor constructs as determined from mNG fluorescence at 530 nm upon excitation with 480 nm light (Fig. 3E, F).

We then determined the BRET ratio of the sensor constructs in live cells under different M$^{pro}$ coexpression conditions as a ratio of emission at 533 nm and 467 nm. Basal BRET ratio of the short and long sensors were found to be 2.17 ± 0.04 vs 1.71 ± 0.20 (mean ± standard deviation; $N = 3$ each; independent experiments performed in triplicates; $p < 0.0001$), respectively, indicating that the additional 6 residues in the long sensor resulted in a 21 (±9)% decrease in the BRET ratio. Coexpression of the wild type M$^{pro}$ resulted in a significant decrease in the BRET ratio while no significant decrease was observed in the presence of the C145A mutant M$^{pro}$ (Fig. 3G). Importantly, both the short and the long sensor expressing cells showed 90.8 (±6.9)% and 92.5

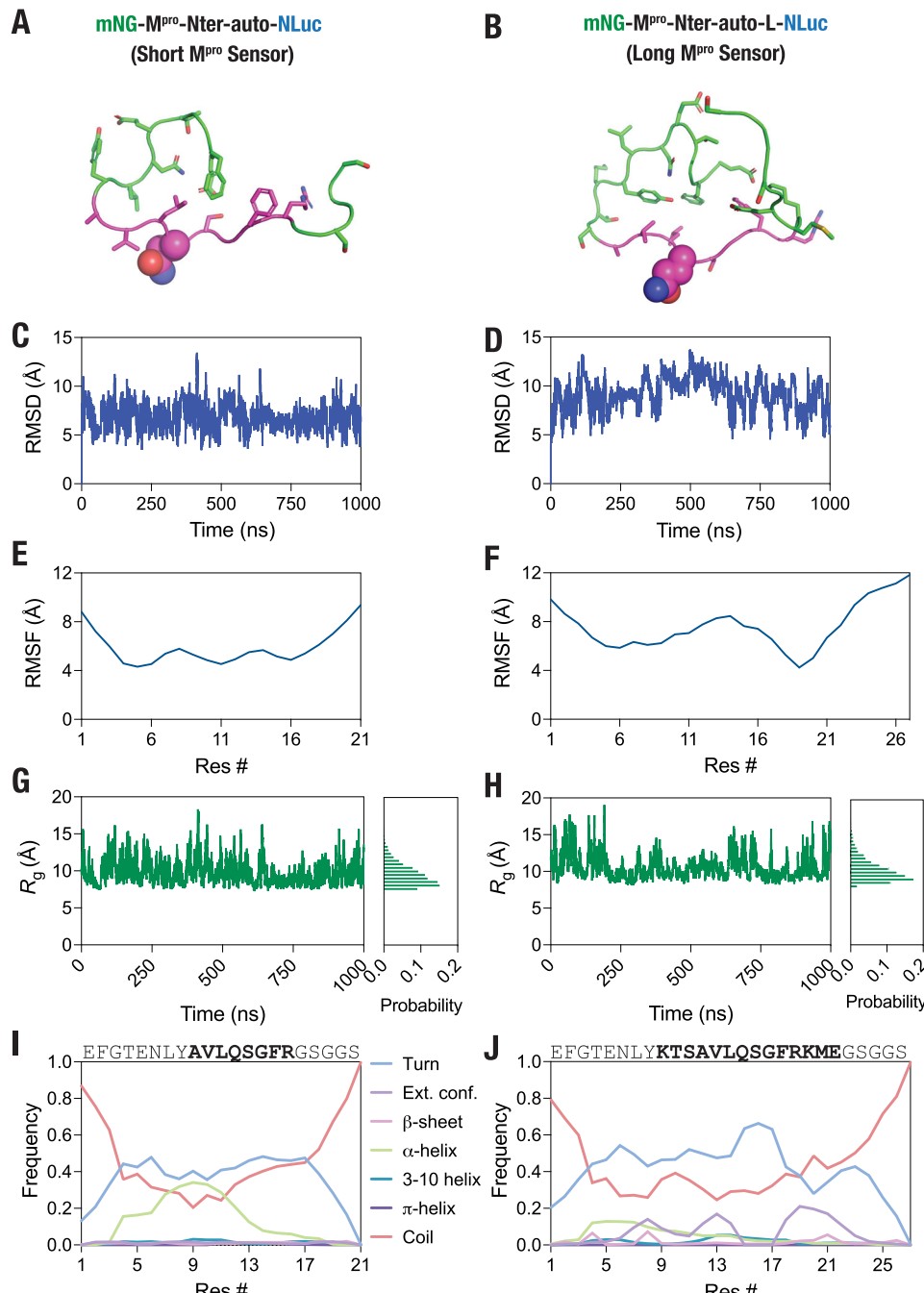

**Fig. 2 M$^{pro}$ N-terminal autocleavage peptide is flexible. A, B** Cartoon representation of the M$^{pro}$-Nter-auto (short; **A**) and M$^{pro}$-Nter-auto-L (long; **B**) peptide structures modeled using the peptide substrate crystallized with H41A mutant SARS-CoV M$^{pro}$ (PDB: 2Q6G[68]). **C, D** Graph showing backbone (Cα) root-mean-square deviation (RMSD) values of M$^{pro}$-Nter-auto (short; **C**) and M$^{pro}$-Nter-auto-L (long; **D**) peptide obtained from 1 μs of GaMD simulations. **E, F** Graph showing backbone (Cα) root-mean-square fluctuation (RMSF) values of M$^{pro}$-Nter-auto (short; **E**) and M$^{pro}$-Nter-auto-L (long; **F**) peptides. **G, H** Graph showing radius of gyration ($R_g$) of the M$^{pro}$-Nter-auto (short; **G**) and M$^{pro}$-Nter-auto-L (long; **H**) peptides monitored over 1 μs of GaMD simulations. **I, J** Graph showing frequency of the indicated secondary structures formed by the M$^{pro}$-Nter-auto (short; **I**) and M$^{pro}$-Nter-auto-L (long; **J**) peptides over the 1 μs of GaMD simulations. T turn, E extended conformation, B isolated bridge, H α-helix, G 3-10 helix, I π-helix, C coil.

(±4.3)%, respectively, reduction in the BRET ratio in the presence of the wild type M$^{pro}$ (Fig. 3G, inset), suggesting that these sensors will provide a wide dynamic range for monitoring M$^{pro}$ proteolytic cleavage activity in live cells. Importantly, no change in the BRET ratio in cells expressing either of the sensors was observed in presence of the C145A mutant M$^{pro}$ indicating high specificity of the BRET signals of these sensors (Fig. 3G, inset).

In order to confirm that the reductions in BRET observed upon coexpression with M$^{pro}$ is specifically due to the cleavage of the

sensors, we performed western blot analysis of cell lysates prepared from the M$^{pro}$ sensor transfected cells. For this, we utilized the N-terminal His$_6$-tag in the M$^{pro}$ sensor constructs, which will be retained in the N-terminal, mNG protein containing fragment upon proteolytic cleavage, and the C-terminal 2x Strep-tag in the M$^{pro}$ protein for detecting cleavage of the M$^{pro}$ sensor constructs and the expression of M$^{pro}$, respectively. Cells transfected with only the M$^{pro}$ sensor constructs showed a band of the expected molecular weight of ~50 kDa (as predicted from

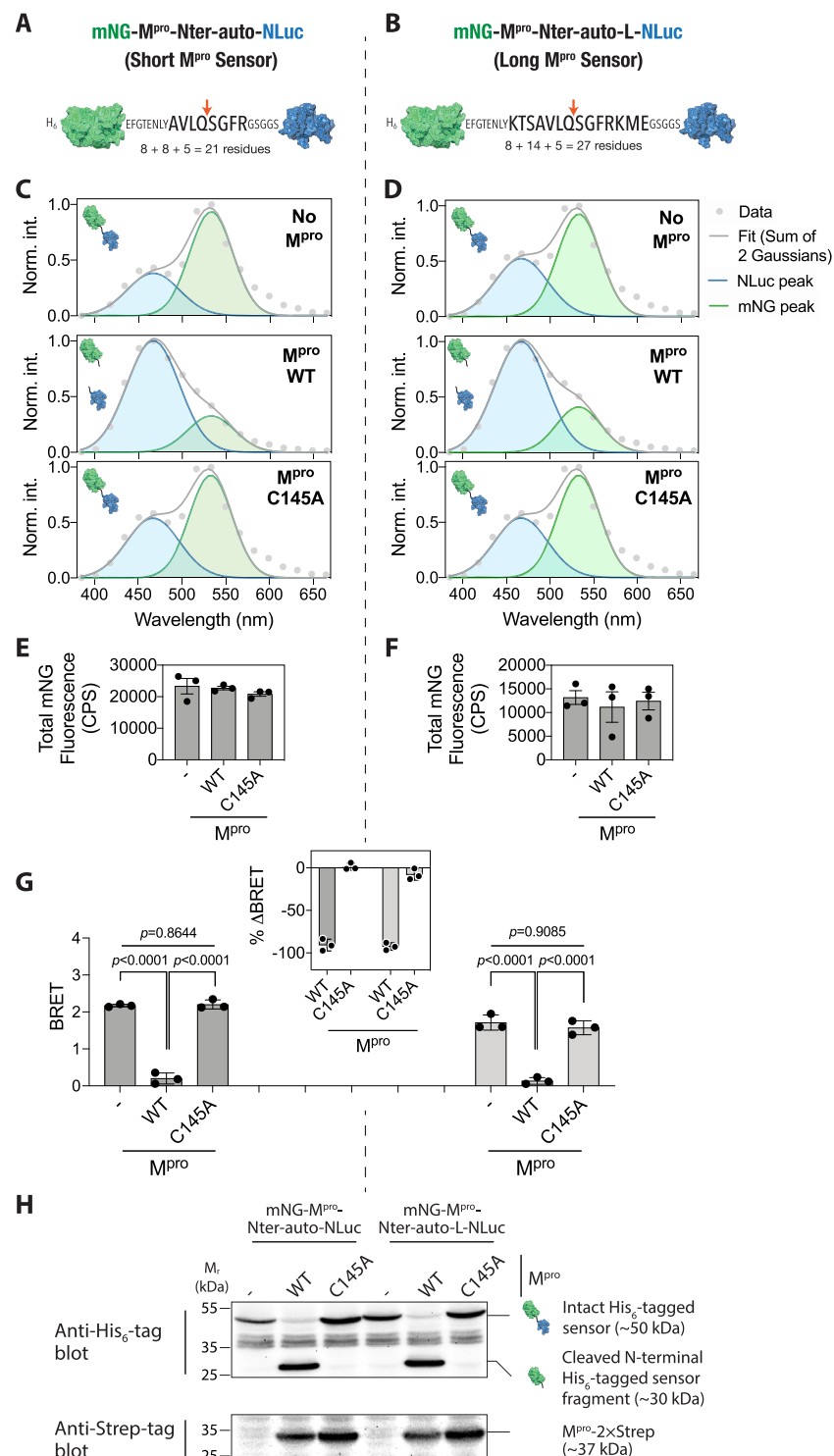

**Fig. 3 Efficient cleavage of the M^pro sensor constructs in live cells. A, B** Schematic showing the M^pro sensor constructs—short (**A**) and long (**B**)—with SARS-CoV-2 M^pro N-terminal autocleavage sequence. **C, D** Graph showing a representative bioluminescence spectra of the short (**C**) and long (**D**) M^pro sensor constructs either in control cells or in cells expressing the WT or C145A mutant M^pro. Data were fit to a two Gaussian model reflecting mNG fluorescence and NLuc bioluminescence peaks. Note the reduction in the mNG peak (533 nm) of both the short and the long sensors when coexpressed with the wild type M^pro while no reduction was observed when coexpressed with the C145A mutant M^pro. **E, F** Graphs showing total mNG fluorescence (measured prior to substrate addition) in cells expressing the short (**E**) and the long (**F**) sensors. **G** Graph showing BRET ratio (ratio emission at 533 and 467 nm) of the short (left side) and the long (right side) M^pro protease activity sensors in either control cells or when coexpressed with the wild type or the C145A mutant M^pro. Note that data for both the short and the long sensor have been plotted on the same y-axis for the ease of comparison. Inset: graph showing percentage change in BRET of the short (left side) and the long (right side) when co-expressed with the wild type or the C145A mutant M^pro. Data shown are mean ± S.D. from three independent experiments, each performed in triplicates. **H** Top panel: anti-His tag blot showing cleavage of the short (left side) and the long (right side) M^pro sensor constructs in either control cells or in cells co-expressing the wild type or the C145A mutant M^pro. Note the release of an approximately 30 kDa, His_6-tagged-mNG fragment in cells expressing the wild type but not in the C145A mutant M^pro. Bottom panel: anti-Strep-tag blot showing expression of the M^pro in the respectively transfected cells.

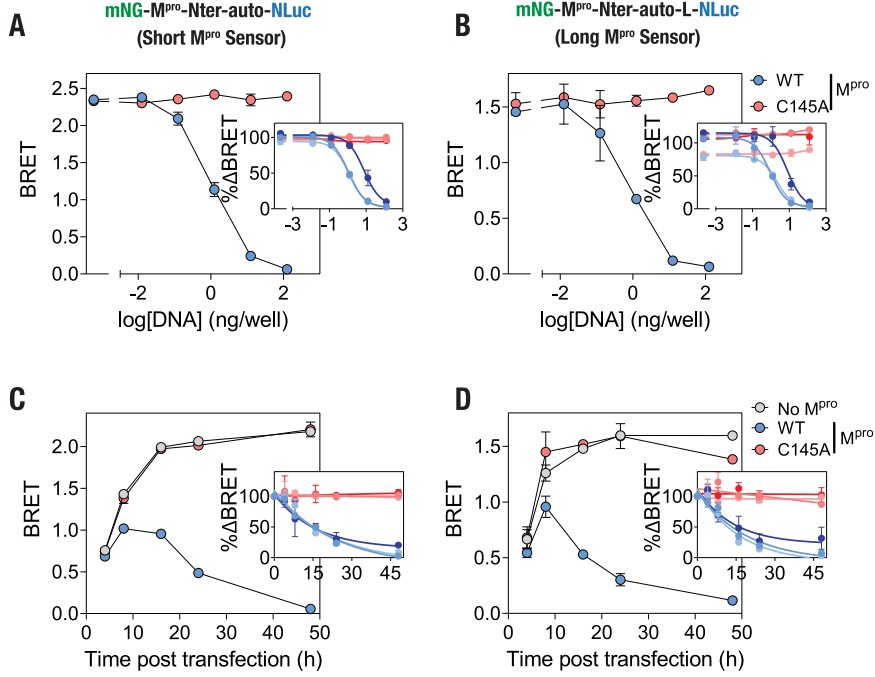

**Fig. 4 $M^{pro}$ DNA concentration- and time-dependent cleavage of the $M^{pro}$ sensors in live cells. A, B** Graph showing BRET ratio of the short (**A**) and the long (**B**) $M^{pro}$ sensors in cells transfected with the indicated amounts of either the wild type or the C145A mutant $M^{pro}$ plasmid DNA. Insets: graph showing percentage decrease in BRET ratio compared to the control cells when transfected with the indicated amounts of the wild type $M^{pro}$ plasmid DNA. Data shown are mean ± S.D. from a representative of three independent experiments, with each experiment performed in triplicates. Data shown in insets are % change in BRET (mean ± S.D) obtained from all three independent experiments and fitted to a sigmoidal dose-response curve. **C, D** Graphs showing the BRET ratio of the short (**C**) and the long (**D**) $M^{pro}$ sensors at the indicated time post transfection in either control cells or cells transfected with the wild type or the C145A mutant $M^{pro}$. Insets: graph showing percentage change in BRET ratio compared to the control cells with time when transfected with the wild-type or mutant $M^{pro}$. Data shown are mean ± S.D. from a representative of three independent experiments, with each experiment performed in triplicates. Data shown in insets are % change in BRET (mean ± S.D) obtained from all three independent experiments and fitted to a one-phase decay curve.

the amino acid sequence of the sensor constructs) (Fig. 3H and Supplementary Fig. 3). Indeed, cells coexpressing the wild type $M^{pro}$, as assessed from the anti-Strep-tag blot, showed a band of ~30 kDa corresponding to the predicted molecular weight of the cleaved N-terminal fragment containing the His$_6$-tag and the mNG protein with a concomitant loss of the full-length sensor construct bands (Fig. 3H). However, cells coexpressing the C145A mutant $M^{pro}$, as assessed from the anti-Strep-tag blot, did not show the cleaved sensor fragments (Fig. 3H). Agreement of these results with the BRET measurements shown above establishes that the reduction in the BRET ratio observed in the presence of the wild type $M^{pro}$ is due to the proteolytic cleavage of the sensor constructs, and therefore, live cell BRET ratio measurements can be reliably used as a measure of $M^{pro}$ proteolytic activity.

**$M^{pro}$ DNA concentration-dependent cleavage of the sensor in live cells.** Having established that the BRET ratio could be used to detect $M^{pro}$ proteolytic activity of the sensor constructs, we aimed to determine the $M^{pro}$ DNA concentration-dependent cleavage of the sensor constructs in live cells. For this, we cotransfected cells with 25 ng/well sensor construct plasmid DNA and a range of $M^{pro}$ plasmid DNA concentrations (0, 0.0125, 0.125, 1.25, 12.5 and 125 ng/well) to gradually increase the number of cells expressing $M^{pro}$ and monitored bioluminescence spectra in adherent cells after 48 h. This revealed a $M^{pro}$ plasmid DNA concentration-dependent shift in the bioluminescence spectra (Supplementary Fig. 4) and a decrease in BRET ratios (Fig. 4A, B) of both the short and the long sensor in the presence of the wild type $M^{pro}$ but not in the presence of the C145A mutant $M^{pro}$.

Discernable decreases in the BRET ratio could be observed at a minimum amount of 1.25 ng/well of $M^{pro}$ plasmid DNA and a maximum decrease in the BRET ratio of ~100% at the highest concentration of 125 ng/well for both sensor constructs (Fig. 4A, B; insets). This analysis has also revealed $EC_{50}$ values are 1.77 ± 1.07 ng/well and 1.85 ± 1.01 ng/well of the Mpro plasmid DNA for the cleavage of the short and the long sensor, respectively. These data demonstrate the functional potency of $M^{pro}$ expressed in these cells.

**Monitoring temporal dynamics of $M^{pro}$ proteolytic activity in live cells.** We then monitored the temporal dynamics of $M^{pro}$ proteolytic activity in live cells using the BRET-based sensors. Towards this, we transfected the cells with the $M^{pro}$ sensor constructs either in the absence or in the presence of the wild type or the C145A mutant $M^{pro}$ plasmid and monitored the bioluminescence spectra from 4 h until 48 h post transfection (Supplementary Fig. 5). Analysis of the bioluminescence spectra obtained from cells expressing either of the $M^{pro}$ sensors indicated a lower BRET ratio after 4 h of transfection, which increased with time and eventually plateaued after 16 h of transfection in the absence of $M^{pro}$ (Fig. 4C, D). Although mNG shows a relatively fast maturation time compared to several other fluorescent proteins[73], these data likely indicate a relatively slower intracellular maturation of mNG compared to NLuc[45,74] (Supplementary Fig. 6). Importantly, a significant decrease in the BRET ratio of cells expressing either of the $M^{pro}$ sensors could be observed in the presence of wild type $M^{pro}$ after 8 h of transfection (Fig. 4C, D). BRET ratio of the cells continued to decrease in

the presence of the wild type $M^{pro}$ until 48 h of transfection while no such decreases were observed in the presence of C145A mutant $M^{pro}$ (Fig. 4C, D; insets). The half-lives of the proteolytic cleavage were found to be $9.76 \pm 3.06$ h and $6.53 \pm 2.88$ h for the short and the long sensor, respectively. These data suggests that the BRET-based $M^{pro}$ sensors developed here could report $M^{pro}$ proteolytic activity as early as 8 h of infection, although this may vary depending on the actual expression of the protease in host cells.

**Comparison of the BRET-based sensors with the FlipGFP-based $M^{pro}$ sensor in live cells.** Having established the monitoring of expression-dependent proteolytic activity of $M^{pro}$ in live cells, we then performed similar experiments with the recently reported FlipGFP-based $M^{pro}$ proteolytic activity reporter[28,29] to compare their performance of the biosensors in reporting $M^{pro}$ proteolytic activity in live cells. For this, we transfected HEK 293T cells with the FlipGFP $M^{pro}$ sensor expression plasmid along with either the WT or C145A $M^{pro}$ expression plasmid and monitored GFP expression in the cells to ascertain conversion of the non-fluorescent protein to a fluorescent one while mCherry expression in the cells was used for detecting transfected cells. Epifluorescence imaging of the cells at different intervals post transfection revealed the appearance of GFP$^+$ cells (transfection of cells ascertained using mCherry fluorescence) in the presence of WT $M^{pro}$ after 24 h of transfection ($67 \pm 7\%$; mean $\pm$ S.D., $n = 2$) while more cells turned GFP$^+$ after 48 h of transfection ($84 \pm 2\%$; mean $\pm$ S.D., $n = 2$) (Supplementary Fig. 7). These data indicate a delayed response of the FlipGFP sensor to $M^{pro}$ proteolytic activity in comparison to the BRET-based sensor. Additionally, a significant number of cells were found to be GFP positive after 48 h ($11 \pm 6\%$; mean $\pm$ S.D., $n = 2$) of FlipGFP transfection in the presence of the C145A mutant $M^{pro}$ (Supplementary Fig. 7). This is contrast to the observations made with the BRET-based sensor in the presence of the mutant $M^{pro}$ (Fig. 4C, D).

**Monitoring pharmacological inhibition of $M^{pro}$ proteolytic activity in live cells.** Finally, we determined the utility of the BRET-based $M^{pro}$ sensors in pharmacological inhibition of $M^{pro}$ proteolytic activity in live cells. Towards this, we cotransfected cells with the $M^{pro}$ sensors and $M^{pro}$ (either WT or C145A mutant) and simultaneously treated the cells with various concentrations of GC376, which has been shown to inhibit $M^{pro}$ in live cells[27,57–59] and determined bioluminescence spectra of the cells after 16 h (Supplementary Fig. 8). A GC376 dose-dependent increase in the BRET ratio of cells coexpressing either the short or the long sensor and the wild type $M^{pro}$ was observed, while no sensor cleavage was observed in the presence of the C145 A mutant $M^{pro}$ (Fig. 5A, B). Percentage proteolytic cleavage activity determined from the BRET ratio indicated an inhibition of $M^{pro}$ starting from a GC376 concentration of 3.33 μM and continued to do so until a concentration of 333 μM (Fig. 5A, B; insets). These data showed that the IC$_{50}$ values for the short sensor is $9.80 \pm 3.84$ μM and that for long sensor is $17.86 \pm 2.14$ μM, which are in a general agreement with the IC$_{50}$ value obtained using the FlipGFP sensor[28] ($5.453 \pm 1.03$ μM; Fig. 5C) and those reported previously[27,28,56–59]. Taken together, these data indicate that the BRET-based $M^{pro}$ proteolytic activity sensors developed here can be utilized for screening antivirals targeted against $M^{pro}$.

**Monitoring $M^{pro}$ proteolytic cleavage activity in vitro.** Having established the utility of the BRET-based $M^{pro}$ sensor in live cell studies, we then focused on determining their utility in vitro using a recombinantly purified $M^{pro}$. For this, we prepared lysates from

HEK 293T expressing either the short or the long $M^{pro}$ sensor construct, incubated equivalent amounts of the lysates with three different concentrations (5 μM, 500 nM, and 50 nM) of the recombinantly purified $M^{pro}$ and monitored BRET following addition of the NLuc substrate (Fig. 6A, B). This assay revealed a $M^{pro}$ concentration-dependent proteolytic processing of the $M^{pro}$ sensors as ascertained from the decreasing BRET ratios. Importantly, the assay also indicated that a minimum of 500 nM of the recombinantly purified $M^{pro}$ is required for a discernable proteolytic cleavage of the sensors as the BRET ratio of the sensors decreased to a lesser extent in the presence of 50 nM $M^{pro}$ protein while a substantially higher rate of cleavage was observed under 500 nM $M^{pro}$. Enzyme kinetics assays performed using a range of the short $M^{pro}$ sensor concentrations 200 nM of $M^{pro}$ revealed a low $k'$ value of $3.09 \pm 0.02$ μM, which is lower than the $K_m$ value of 17[75] or 56 μM[76] determined using a FRET-based peptide substrate, and a Hill coefficient of 1.58 (Supplementary Fig. 9), suggesting a significant cooperativity in the protein. Further, an increase in $M^{pro}$ concentration resulted in a decrease in the Hill coefficient to 1.16 (Supplementary Fig. 9), suggesting a role for protein dimerization in the observed cooperativity in $M^{pro}$[77]. We then performed the assays in the presence of GC376 to determine the pharmacological inhibition of $M^{pro}$ activity in vitro. For this, 500 nM $M^{pro}$ was preincubated with a range of concentrations of GC376 ($10^{-4}$–$10^{-9}$ M) for 30 min at 37 °C and cleavage activity was monitored after addition of lysates prepared from cells expressing either the short or the long $M^{pro}$ sensor. Incubation with GC376 resulted in a decrease in the rate of proteolytic cleavage of both the short and the long $M^{pro}$ sensor (Fig. 6C, D) with IC$_{50}$ values of $73.1 \pm 7.4$ and $86.9 \pm 11.0$ nM for the short and the long sensors, respectively (Fig. 6G, H).

**In vitro assays reveal molecular crowding-mediated increase in $M^{pro}$ proteolytic activity and a decrease in inhibitor efficacy.** We took advantage of the slow rate of the $M^{pro}$ sensor cleavage under 500 nM $M^{pro}$ to determine the effect of molecular crowding on the proteolytic activity of the protein. Molecular crowding in the intracellular environment caused by the presence of soluble and insoluble macromolecules such as proteins, nucleic acids, ribosomes and carbohydrates has been shown to impact both structure and stability of proteins in cells as well as enzyme kinetics[78–83] including a decrease in the activity of hepatitis C virus NS3/4A protease[84] and an increase in the proteolytic activity of SARS-CoV $M^{pro}$[82]. For this, we included 25% (weight/volume; w/v) of 8000 Da polyethylene glycol (PEG 8000), which is a non-toxic, hydrophilic polyether that serves as a crowding agent and has been extensively utilized to mimic molecular crowding in vitro[85,86], in the assays and monitored cleavage of the $M^{pro}$ sensors with 500 nM $M^{pro}$ under varying concentrations of GC376. Inclusion of 25% PEG 8000 resulted in a substantial increase in the rate of proteolytic cleavage of the $M^{pro}$ sensors in the absence of GC376 (Fig. 6E, F). This data suggests that molecular crowding caused by PEG 8000 is likely effective in causing increased dimerization of $M^{pro}$, a feature critical for its catalytic activity, through an increase in the effective concentration of the protein due to excluded volume effects, and thus increases the rate of proteolytic cleavage of the $M^{pro}$ sensor. Importantly, while GC376 could inhibit $M^{pro}$ activity, the IC$_{50}$ values as obtained from BRET ratios after 2 h of incubation with both the short as well as the long sensor indicated a large shift (IC$_{50}$ values of $2623 \pm 760$ and $10,260 \pm 3280$ nM, respectively) (Fig. 6G, H). These IC$_{50}$ values are in agreement with those obtained from live cell assays and suggest that molecular crowding prevalent in living cells might activate $M^{pro}$ and impact the inhibitory potential of $M^{pro}$ inhibitors developed to treat COVID-19.

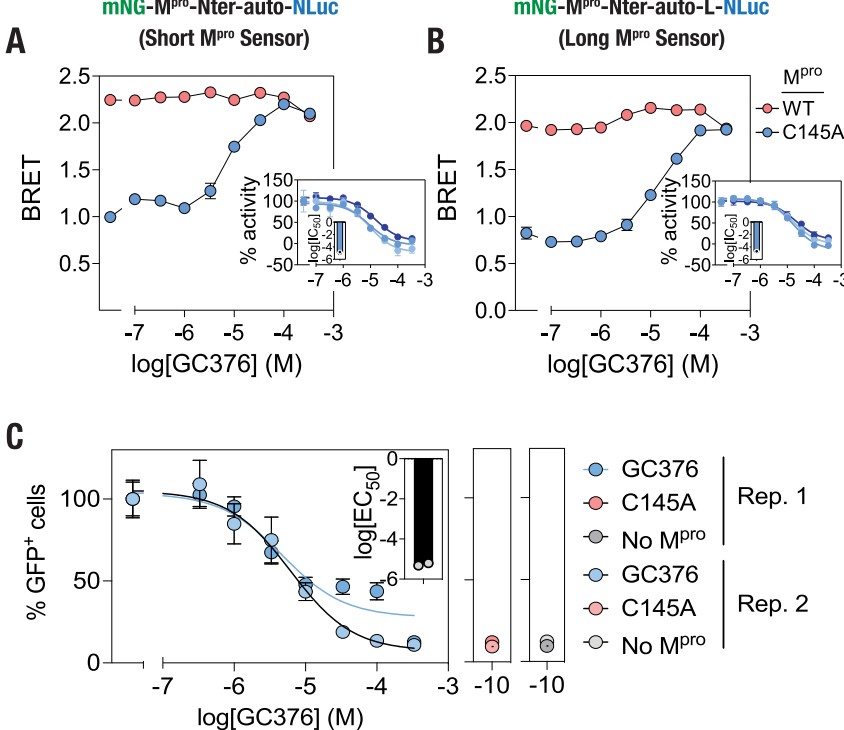

**Fig. 5 M^pro inhibition monitored in live cells using the BRET-based M^pro sensors. A, B** Graphs showing GC376-mediated inhibition of M^pro proteolytic cleavage of the short (**A**) and long (**B**) M^pro sensor in live cells. Data shown are mean ± S.D. from a representative of three independent experiments, with each experiment performed in triplicates. Insets: graphs showing percentage proteolytic activity of WT M^pro in live cells (mean ± S.D.) and fitted to sigmoidal dose–response curve from three independent experiments. Bar graphs in the insets show log(IC_{50}) obtained from the three experiments. **C** Graph showing % GFP^+ cells expressing the FlipGFP-based M^pro sensor and wild-type M^pro at the indicated concentrations of GC376. Inset, graph showing log(IC_{50}) values (mean of two independent experiments). Outsets, graphs showing % GFP^+ cells expressing the C145A mutant M^pro (left) or in the absence of M^pro (right). Data shown are mean of two independent experiments, with each experiment performed in triplicates.

## Conclusion

To conclude, we have developed genetically encoded, BRET-based M^pro protease activity sensors for use in live cells as well as in vitro assays and have validated their utility for antiviral drug discovery using GC376 as a proof of principle. The use of BRET, with NLuc as the bioluminescence donor and mNG as the resonance energy acceptor, enabled highly sensitive detection of M^pro proteolytic activity in live cells. Additionally, the sensors developed here did not show any cleavage, either in the absence of M^pro or in the presence of the catalytically dead, C145A mutant M^pro, thus, displaying high specificity. The BRET-based sensor allowed detection of molecular crowding-mediated increased activity of M^pro enabling the discovery of reduced inhibitory efficacy of GC376 under a crowded condition. We believe that these sensors will find utility in both detecting active SARS-CoV-2 infection as well in screening antivirals developed for targeting M^pro proteolytic cleavage activity in live cells. Additionally, they can be utilized for determining effects of genetic variation in the M^pro amino acid sequence that may arise during the evolution of the virus.

## Methods

**M^pro N-terminal autocleavage sequence analysis**. A total of 1984 sequences for the SARS-CoV-2 pp1a polyprotein available at the NCBI Virus database (https://www.ncbi.nlm.nih.gov/genome/viruses/) were downloaded and aligned using MAFFT server (https://mafft.cbrc.jp/alignment/server/)[87,88]. The aligned sequences of the pp1a polyprotein were analyzed for the conservation of the M^pro N-terminal autocleavage positions (AVLQSGFR).

**Structural modeling of M^pro N-terminal autocleavage peptide sequences**. The crystal structure of the N-terminal peptide substrate complexed with SARS-CoV main protease H41A mutant (PDB: 2Q6G[68], Chain D, aa seq: TSAVLQSGFRK) was used as a template for generating the 3D models for the short and long M^pro

cleavage peptides, including the linker region, of the Mpro sensor (short cleavage peptide aa seq: EFGTENLYAVLQSGFRGSGGS, long cleavage peptide aa seq: EFGTENLYKTSAVLQSGFRKMEGSGGS). Models were generated using MODELLER (10.1 release, Mar. 18, 2021)[71]. Briefly, the short and long sequences were aligned with the template in PIR format. For each peptide, 100 models were initially generated using "Automodel" function and "very-slow" MD refining mode. Scoring functions such as modpdf, DOPE, and GA34, were used to assess the generated models. The model with the lowest DOPE score was further refined by loop modeling using very-slow loop MD refining mode to generate 100 refined models. The same scoring functions were used to assess the refined models. The stereochemical quality of the final model was assessed with PROCHECK[89].

**Molecular dynamics simulation**. To neutralize the positive and negative charges on the peptide's termini, the N- and C-termini were capped with *N*-acetyl and *N*-methyl amide capping groups, respectively. Topology and parameter files were generated using CHARMM-GUI webserver[90]. The biomolecular simulation systems included the peptide model, with all hydrogens added, solvated in TIP3P (transferable intermolecular potential with 3 points)[91] cubic water box with 10 Å minimum distance between edge of box and any of the peptide atoms. Charges were neutralized by adding 0.15 M NaCl to the solvated system. The total number of atoms was 15480 and 18233 for the short and long peptide simulation systems, respectively. In silico molecular dynamics simulations were performed using Nanoscale Molecular Dynamics (NAMD) software[72] version 2.13 with the CHARMM36(m) force field[92]. A 2 fs time-step of integration was set for all simulations performed. First, energy minimization was performed on each system for 1000 steps (2 ps). Following energy minimization, the system was slowly heated from 60 to 310 at 1 K interval to reach the 310 K equilibrium temperature using a temperature ramp that runs 500 steps after each temperature increment. Following thermalization, temperature was maintained at 310 K using Langevin temperature control and at 1.0 atm using Nose–Hoover Langevin piston pressure control[93,94]. The system was then equilibrated with 500,000 steps (1 ns) using Periodic Boundary Conditions. The NAMD output structure was then used as an input for Gaussian accelerated molecular dynamics (GaMD) simulation utilizing the integrated GaMD module in NAMD and its default parameters[69,70], which included 2 ns of conventional molecular dynamics (cMD) equilibration run in GaMD, to collect potential statistics required for calculating the GaMD acceleration parameters, and another 50 ns equilibration run in GaMD after adding the boost

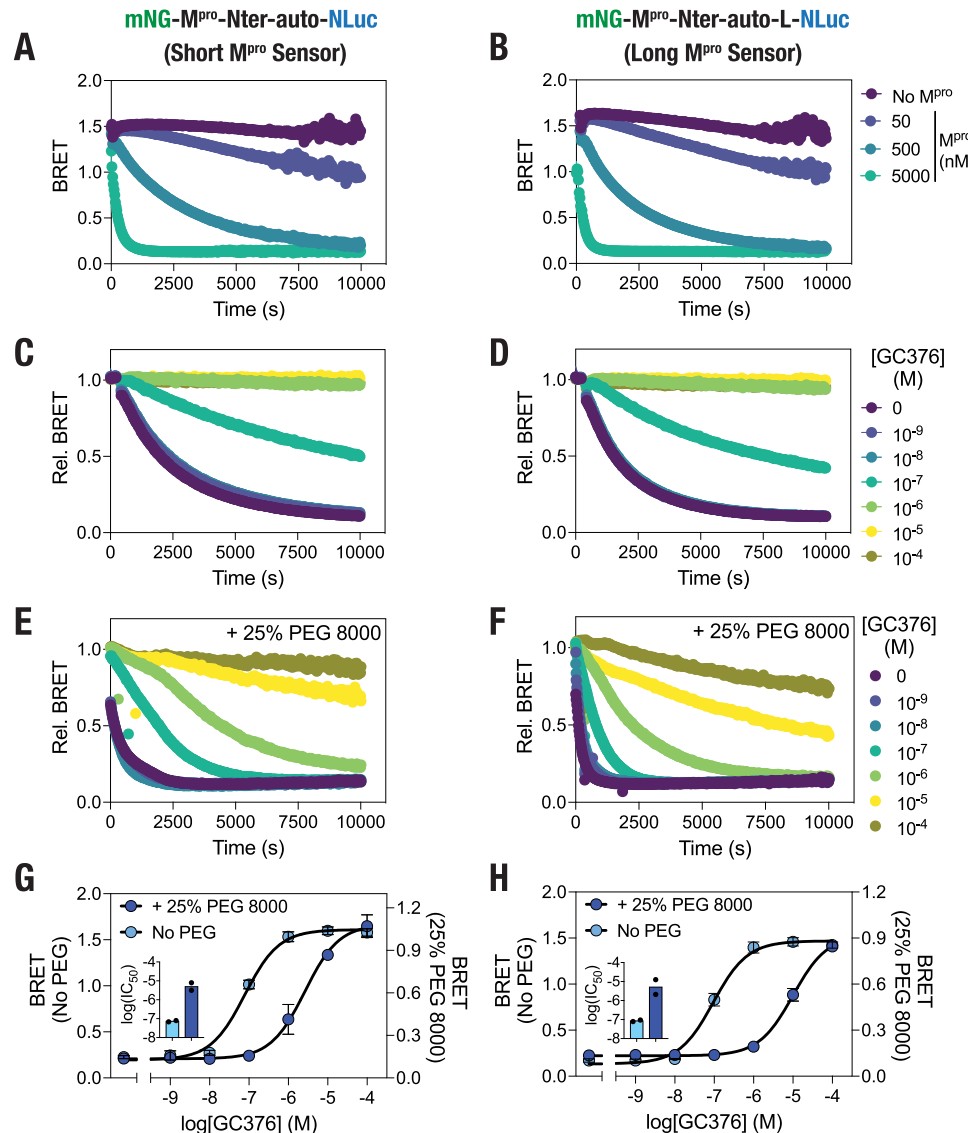

**Fig. 6 Molecular crowding-mediated increase in M^pro proteolytic activity and decrease in GC376 potency. A**, **B** Graph showing in vitro proteolytic cleavage kinetics of the short (**A**) and the long (**B**) M^pro biosensor under the indicated concentrations of the recombinantly purified M^pro protein. **C**, **D** Graphs showing GC376-mediated inhibition of SARS-CoV M^pro proteolytic cleavage of the short (**C**) or the long (**D**) M^pro sensors. **E**, **F** Graphs showing GC376-mediated inhibition of M^pro proteolytic cleavage of the short (**E**) or the long (**F**) M^pro sensors in the presence of 25% (w/v) of PEG 8000. **G**, **H** Graph showing concentration-dependent inhibition of M^pro determined using either the short (**G**) or the long (**H**) M^pro sensor in the absence or in the presence of 25% (w/v) of PEG 8000. Inset in **G**, **H**: graphs showing log(IC$_{50}$) values of GC376 in the absence and presence of 25% (w/v) of PEG 8000. Data shown are mean obtained from assays repeated twice, with each experiment performed in duplicates.

potential[70,95], and finally GaMD production runs for 1000 ns. Both equilibration steps in GaMD were preceded by 0.4 ns preparatory runs. All GaMD simulations were run at the "dual-boost" level by setting the reference energy to the lower bound, i.e., $E = V_{max}$[96]. One boost potential is applied to the dihedral energetic term and the other to the total potential energetic term. The details for calculating the boost potentials including the equations used have been described previously[69,96,97]. The upper limits of standard deviation (SD) of the dihedral and total potential boosts in GaMD were set to 6.0 kcal/mol. All GaMD simulations were performed using similar and constant temperature and pressure parameters. For all simulations, short-range non-bonded interactions were defined at 12 Å cut-off with 10 Å switching distance, while Particle-mesh Ewald (PME) scheme was used to handle long-range electrostatic interactions at 1 Å PME grid spacing. Trajectory frames were saved every 10,000 steps (20 ps) and trajectory analysis was performed using the available tools in Visual Molecular Dynamics (VMD) software[98]. Trajectory movies were compiled based on 1000 frames using Videomach (http://gromada.com/videomach/) to generate 41 s movies (24 fps) in AVI format.

**M^pro BRET sensor plasmid construct generation**. The BRET-based M^pro activity sensors were developed based on previously reported M^pro N-terminal auto-cleavage peptides, namely AVLQSGFR[4] (nucleotide sequence 5' GCA GTG CTC

CAA AGC GGA TTT CGC 3') and KTSAVLQSGFRKME[5,26] (nucleotide sequence 5' AAA ACG AGT GCC GTA TTG CAG AGT GGG TTT CGG AAA ATG GAA 3'), referred to as mNG-M^pro-Nter-auto-NLuc and mNG-M^pro-Nter-auto-L-NLuc, respectively. For this, fragments BstXI-mNG-M^pro-Nter-auto-NLuc-XhoI and BstXI-mNG-M^pro-Nter-auto-L-NLuc-XhoI were synthesized (Integrated DNA Technologies, IDT; Iowa, USA) and inserted into pIDTSmart (Kan) vectors to generate the plasmid constructs pIDT-mNG-M^pro-Nter-auto-NLuc and pIDT-mNG-M^pro-Nter-auto-L-NLuc, respectively. Both vectors were transformed into *E. coli* for amplification and purified using Qiagen mini-prep kit. Restriction enzymes *Bst*XI and *Xho*I were used to excise the two DNA fragments of interests from entry clones pIDT-mNG-M^pro-Nter-auto-NLuc and pIDT-mNG-M^pro-Nter-auto-L-NLuc and ligated into similarly digested destination plasmid pmNeonGreen-DEVD-NLuc [Addgene: 98287][46] and further confirmed by Sanger sequencing. One Shot TOP10 Competent *E. coli* cells were transformed with 2 μL of the ligation reaction and plated on LB agar plates with 100 μg/mL ampicillin. Sequences were confirmed by Sanger sequencing[99] using a pair of forward and reverse primers, 5' GCACAGCCAGAACCACATATACCTT 3' and 5' CACCACCTTGAAGATCTT CTCGATCT 3', respectively. For bacterial expression and purification of the M^pro sensor, the mNG-M^pro-Nter-auto-NLuc plasmid construct was digested with *Hind*III and *Xho*I and the mNG-M^pro-Nter-auto-NLuc fragment was subcloned into similarly digested pET-28b(+) plasmid.

**Cell culture and transfection**. All experiments reported in the manuscript were performed with HEK 293T cells, which were grown in Dulbecco's Modified Eagle Medium (DMEM) supplemented with 10% fetal bovine serum, and 1% penicillin–streptomycin and grown at 37 °C in 5% $CO_2$[18,35–37,100–104]. Transfections were performed with polyethyleneimine (PEI) lipid according to manufacturers' protocol. Briefly, HEK 293T cells were seeded onto 96-well white plates before 24 h of transfection. The plasmid DNA (sensor and $M^{pro}$), Opti-MEM (Invitrogen; 31985088) and 1.25 μg/well of PEI lipid (Sigma-Aldrich; 408727-100 mL) were combined using pipetting and incubated at room temperature for 30 min before being added to cells by droplet. The PEI stock solution of 2 mg/mL was prepared by diluting in sterile Milli-Q water and stored at −80 °C.

**Live cell, BRET-based $M^{pro}$ proteolytic cleavage activity assays**. Live cell $M^{pro}$ proteolytic cleavage activity assays were performed by co-transfecting HEK 293 T cells with either the pmNG-$M^{pro}$-Nter-auto-NLuc or the pmNG-$M^{pro}$-Nter-auto-L-NLuc $M^{pro}$ sensor plasmid constructs along with either pLVX-EF1alpha-SARS-CoV-2-nsp5-2xStrep-IRES-Puro ($M^{pro}$ WT) (a gift from Nevan Krogan (Addgene plasmid # 141370; http://n2t.net/addgene:141370; RRID:Addgene_141370)[54] or pLVX-EF1alpha-SARS-CoV-2-nsp5-C145A-2xStrep-IRES-Puro (C145A mutant $M^{pro}$) plasmid (a gift from Nevan Krogan (Addgene plasmid # 141371; http://n2t.net/addgene:141371; RRID:Addgene_141371)[54] in 96-well white flat bottom plates (Nunc; 136101). For dose-response experiments, a control plasmid (a pcDNA3.1-based plasmid not expressing $M^{pro}$) is also co-transfected to maintain the amount of plasmid DNA in transfection constant. In case of time-course experiments with either the short or the long sensor, control cells (No $M^{pro}$) were transfected with the control plasmid while the wild type (WT) and the C145A mutant $M^{pro}$ cells were transfected with the respective $M^{pro}$ expressing plasmid DNA. The time-course experiments were carried out at 1:5 sensor-to-protease plasmid DNA ratio. Post 48 h (or as indicated in the time course experiments) of transfection, BRET measurements were performed by the addition of furimazine (Promega, WI, USA) at a dilution of 1:200. In time-course experiments, BRET was measured at the indicated time points. Experiments were performed in triplicates and repeated a minimum of three times.

**Western blot analysis**. HEK 293T cells co-transfected with the $M^{pro}$ sensor and the $M^{pro}$ (wild-type or mutant) plasmids were lysed in 200 μL of 2× Laemmli sample buffer (50 mM Tris-Cl pH 6.8, 1.6% SDS, 8% glycerol, 4% β-mercaptoethanol, and 0.04% bromophenol blue) (heated to 85 °C and sonicated prior to addition). Equal volumes of the cell lysates (30 μL) were separated by 10% SDS-PAGE using running buffer (25 mM Tris, 192 mM glycine, 0.1% SDS) at a constant voltage of 100 V for 1.5 h following which proteins were transferred onto PVDF (polyvinylidene fluoride) membranes. Membranes were blocked in Tris-buffered saline containing 0.1% Tween-20 (TBS-T) with skimmed milk (5%) for 1 h at room temperature. Blots were incubated either with anti-His antibody (6×-His Tag Monoclonal Antibody (HIS.H8), Alexa Fluor 488; ThermoFisher Scientific-MA1-21315-A488; 1:5000) or with anti-Strep-tag mouse monoclonal antibody (anti-Strep-tag mouse monoclonal, C23.21; PROGEN- 910STR; 1:5000) overnight at 4 °C in dilution buffer (TBS-T containing 5% bovine serum albumin (BSA). Secondary anti-mouse IgG HRP (Anti-Mouse Ig:HRP Donkey pAb; ECM biosciences- MS3001; 1:10,000 diluted in TBS-T) was used to detect $M^{pro}$ and the cleaved $M^{pro}$ sensor proteins.

**Live cell $M^{pro}$ proteolytic cleavage inhibitor assay**. HEK 293T cells were co-transfected with either pmNG-$M^{pro}$-Nter-auto-NLuc or pmNG-$M^{pro}$-Nter-auto-L-NLuc plasmid along with either pLVX-EF1alpha-SARS-CoV-2-nsp5-2xStrep-IRES-Puro ($M^{pro}$ WT) (Addgene plasmid # 141370) or pLVX-EF1alpha-SARS-CoV-2-nsp5-C145A-2xStrep-IRES-Puro ($M^{pro}$ C145A) (Addgene plasmid # 141371) plasmid in 96-well white flat bottom plates at a ratio of 1:5 of sensor-to-protease plasmid DNA ratio. Transfected cells were concomitantly treated with a range of GC376 (GC376 Sodium; AOBIOUS—AOB36447; stock solution prepared in 50% DMSO at a concentration of 10 mM) concentrations. After 16 h of incubation with the inhibitor, BRET measurements were performed by the addition of furimazine (Promega, Wisconsin, USA) at a dilution of 1:200. The percentage activity was calculated by normalizing the BRET ratio with the negative control (No $M^{pro}$). Three independent experiments were performed in triplicates for each sensor construct.

**Live cell, FlipGFP-based $M^{pro}$ proteolytic assay**. For live cell FlipGFP-based $M^{pro}$ proteolytic activity assays, HEK 293T cells were seeded onto 24-well plates and co-transfected with the FlipGFP sensor plasmid (pcDNA3 FlipGFP($M^{pro}$) T2A mCherry; a gift from Xiaokun Shu; Addgene plasmid # 163078)[29] and either the WT or the C145A mutant $M^{pro}$ expressing plasmid DNA using PEI lipid after 24 h of cell seeding. For transfection, cells were imaged using a EVOS FL microscope (Life Technologies; ×4 objective) at the indicated time in the red (to monitor mCherry expression to determine transfected cells) and the green (to monitor conversion of non-fluorescent FlipGFP into the fluorescent GFP form after $M^{pro}$-mediated cleavage) channels. Images were analyzed for percentage GFP positive (GFP$^+$) cells, number of transfected cells and total number of analyzed cells for each time point using Fiji[105]. The ImageJ macro script used for the analysis is provided in the Supplementary Methods. For determining number of GFP$^+$ cells,

GFP intensities obtained for each cell was background corrected and threshold was applied.

**Cell lysate preparation for in vitro BRET assays**. To prepare cell lysates containing the $M^{pro}$ sensors, HEK 293T cells were transfected with either the mNG-$M^{pro}$-Nter-auto-NLuc or the mNG-$M^{pro}$-Nter-auto-L-NLuc $M^{pro}$ BRET sensor and washed with chilled Dulbecco's Phosphate-Buffered Saline (DPBS) 48 h post transfection. Cells were lysed in a buffer containing 50 mM HEPES (pH 7.5), 50 mM NaCl, 0.1% Triton-X 100, 1 mM dithiothreitol (DTT) & 1 mM ethylene-diamine tetraacetic acid (EDTA)[60] on ice. Cell lysates were collected in a 1.5 mL Eppendorf tube and centrifuged at 4 °C for 1 h at 14,000 rotations per min (RPM) following which supernatant were collected and stored at −80 °C until further usage.

**In vitro, BRET-based $M^{pro}$ proteolytic cleavage activity assays**. In vitro BRET-based $M^{pro}$ proteolytic cleavage activity assays were performed by incubating cell lysates containing the short, BRET-based $M^{pro}$ sensor with different concentrations (0.5, 5, 50, and 500 nM) of recombinantly purified SARS-CoV $M^{pro}$ (SARS coronavirus, 3CL Protease, Recombinant from E. coli; NR-700; BEI Resources, NIAID, NIH; stock solution of the protein was prepared by dissolving the lyophilized protein in Tris-buffered saline (TBS) containing 10% glycerol) at a concentration of 50 μM and BRET monitored through luminescence scans. The effect of molecular crowding was monitored by incubating the sensor and the protease in the absence or presence of 25% (w/v) of PEG 8000 (Sigma-Aldrich). GC376 (GC376 Sodium; AOBIOUS—AOB36447; stock solution prepared in 50% DMSO at a concentration of 10 mM) inhibition of $M^{pro}$ (500 nM) activity was monitored under a range of the inhibitor concentrations in the absence or presence of 25% (w/v) PEG 8000. BRET measurements were performed at 37 °C by the addition of furimazine (Promega, Wisconsin, USA) at a dilution of 1:200. The bioluminescence (467 nm) and fluorescence (533 nm) readings were recorded using Tecan SPARK multimode microplate reader and used to calculate the BRET ratios (533 nm/467 nm). Total mNG fluorescence in cell lysates containing the short, BRET-based $M^{pro}$ sensor was measured by exciting the samples at 480 nm and emission acquired at a wavelength of 530 nm.

**BRET and fluorescence measurements**. BRET measurements were performed using a Tecan SPARK® multimode microplate reader. Bioluminescence spectral scan was performed from 380 to 664 nm wavelengths with an acquisition time of 400 ms for each wavelength to determine relative emissions from NLuc (donor) and mNG (acceptor) and quantify BRET, which is expressed as a ratio of emissions at 533 and 467 nm. In some experiments, BRET measurements were performed by measuring emission only at 533 and 467 nm. Percentage changes in BRET were determined after subtracting a background BRET ratio of 0.2 determined similarly using a recombinantly purified NLuc protein[106]. Total mNG fluorescence in the sensor expressing cells was measured by exciting the samples at 480 nm and emission acquired at a wavelength of 530 nm.

**Data analysis and figure preparation**. GraphPad Prism (version 9 for macOS, GraphPad Software, La Jolla, CA, USA; www.graphpad.com), in combination with Microsoft Excel, was used for data analysis and graph preparation. Figures were assembled using Adobe Illustrator.

**Reporting summary**. Further information on research design is available in the Nature Research Reporting Summary linked to this article.

## Data availability
All the relevant data of this study are available within this paper and the Supplementary Information file.

## Code availability
An ImageJ/Fiji script written for multi-channel fluorescence intensity determination is included in the Supplementary Information file.

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

## Acknowledgements

This work is supported by an internal funding from the College of Health & Life Sciences, Hamad Bin Khalifa University, a member of the Qatar Foundation. A.M.G. is supported by a postdoctoral fellowship and W.S.A., S.R., A.F., and S.M.N.U are supported by scholarships from the College of Health & Life Sciences, Hamad Bin Khalifa University, a member of the Qatar Foundation. Some of the computational research work reported in the manuscript were performed using high-performance computer resources and services provided by the Research Computing group in Texas A&M University at Qatar. Research Computing is funded by the Qatar Foundation for Education, Science and Community Development (http://www.qf.org.qa).

## Author contributions

K.H.B. conceived the experiments. A.M.G., W.S.A., S.R., A.F., S.M.N.U., M.A., and K.H.B. performed experiments, analyzed data, prepared figures, and wrote the manuscript. All authors reviewed the manuscript.

## Competing interests

K.H.B. and A.M.G. are inventors on a US Provisional Patent application number 63/275217, assigned to Qatar Foundation for Education, Science and Community Development, entitled "BRET-based coronavirus M$^{pro}$ protease sensor and uses thereof" relating to the development and uses of the M$^{pro}$ sensor. The other authors declare no competing interests.
