## [Peer Review File · Communications Chemistry]

Reviewers' comments:

Reviewer #1 (Remarks to the Author):

Anupriya et al report on the construction of a BRET-based sensor for measuring SARS-CoV-2 protease inhibition. This is a therapeutic target for Covid-19 treatments, and the creation of rapid/easy detection assays for detecting inhibition have value for the current pandemic and will likely be valuable for other present or future viruses. The paper is well presented and the authors robustly characterize their sensor.

Figure 3G: The inset blocks a portion of the y-axis and its labelling. Please move it to make the axis clear.

Figure 3: The authors state: "Data shown are mean \pm S.D. from a representative of independent experiments performed multiple times". However, all data should be shown. The methods suggest two independent replicates each containing 3 technical replicates, and the presence of only 3 data points looks like only one set of technical replicates were graphed.

Figures 4-7 are likewise unclear on the number of replicates being represented and appear to be a subset of the data collected.

Lines 354-428: Perhaps place this in the supplementary materials. It seems to be validating the authors construct choice, but does not add much value considering the authors rigorously tested both experimentally.

Lines 489 to 531: This experiment is unclear to me. The methods suggest an empty "filler plasmid" was also transformed. This sounds like there will be cells transformed with the filler plasmid rather than Mpro, which isn't really a dose-response of Mpro on the reporter. How do the authors know this is a dose response of Mpro rather than just having more cells transformed with the filler plasmid instead of Mpro?

Lines 33-36, 124-126, 594-611, 740-741: I think the present authors did not perform enough robust experiments to make these claims of increased specificity and sensitivity. The present authors performed one experiment and showed the FlipGFP-based sensor exhibits more background than their one. Background is not that important here because the signal to noise ratio and detection limit are more important for determining specificity or sensitivity. In fact, the opposite appears to be true. The authors of the FlipGFP paper show an IC₅₀ of 5.5 μ M with GC376 (See figure 3D), whereas the present paper found a much higher IC₅₀ of 127 and 194 μ M when testing their sensor with GC376.

Figure 1: The mNG and NLuc on top of the proteins is very hard to read, and the authors might want to give a quick explanation of these in the figure legend.

The referencing needs to be checked in this paper. For instance, should the reference at Line 597 be 29 instead of 30?

I like that the authors included the protein sequences in the supplementary materials. Could they please add DNA sequences as well? Expression can be dependent on the exact DNA sequence and would increase the reproducibility of their work.

Reviewer #2 (Remarks to the Author):

This manuscript reported the design of Bioluminescence Resonance Energy Transfer (BRET)-based

assay for the SARS-CoV-2 main protease (Mpro). Both the shorter and longer substrates were examined, which gave similar results. The ratio of reporter plasmid to the Mpro, as well as the induction time, were optimized. Flip-GFP assay was performed in parallel. It was found that the BRET assay was more sensitive and specific than the Flip-GFP assay. The optimized BRET assay was used to test Mpro inhibitor GC-376 and found to have much weaker potency than the values reported in the literature. The BRET assay was also performed in cell culture, and addition of 25% PEG 20K increased the proteolytic activity of Mpro and decreased the potency of GC-376. It was therefore claimed that Mpro is more active in the crowded environment of an infected host cell compared to in vitro conditions, thus requiring higher drug concentration for complete inhibition. Highlights of this study including the detailed assay optimization and vigorous assay calibration using the C145A dead mutants, the direct comparison with the Flip-GFP assay, and the molecular crowding experiment. To further strength the conclusions, the authors might consider the following suggestions:

1 "This is especially relevant given that the binding of the peptide substrate has been reported to allosterically activate the SARS-CoV-1 Mpro dimer."

Comment: reference should be given.

2. Is there any internal control for the BRET assay to normalize the transfection efficiency? In the Flip-GFP assay, mCherry is the internal control.

3. "Additionally, a significant number of cells were found to be GFP positive after 24 h ($9 \pm 1\%$) and 48 h ($20 \pm 1\%$) of FlipGFP transfection in the presence of the C145A mutant Mpro (Fig. 5C,D). This is contrast to the observations made with the BRET-based sensor in the presence of the mutant Mpro (Fig. 4C,D)."

Comment: the background GFP signal from the Flip-GFP assay might be a result of cleavage by the host proteases. If this the case, the BRET assay should have similar background signal as both assays contain the same substrate. Is there any explanation why the BRET assay has less leakage signal?

4. "The lower efficacy of GC376 observed here compared to previous reports perhaps indicates a cell type- or Mpro 646 expression dependent effect."

Comment: HEK 293T cells were used in both the BRET and Flip-GFP assay, so the above statement does not hold. For direct comparison, the author should also determine the EC50 of GC-376 in the Flip-GFP assay. This is to rule out the possibility of incorrect drug concentration or the different cell type used in this study from the ones reported in the literature. Another possibility might be the drug efflux pump P-gp and GC-376 is a known substrate of P-gp (ACS Infect. Dis. 2021, 7, 3, 586–597). However, this is unlikely as 293T is not known to have high levels of P-gp.

5. "Together, these data indicate that Mpro could be more active in the crowded environment of an infected host cell compared to in vitro conditions, and may require higher concentrations of pharmacological inhibitors for effective inhibitions of its catalytic activity than those determined from in vitro assays."

Comment: To provide additional evidence for this conclusion, the authors should repeat the FRET assay with and without 25% PEG 20K. In addition, the lack of direct correlation between the results of in vitro assay and the cell-based assay might due to many factors including cell membrane permeability, drug efflux, protein binding, off-target effect, metabolism and etc.

Response to Reviewer's comments:

Reviewer #1

Anupriya et al report on the construction of a BRET-based sensor for measuring SARS-CoV-2 protease inhibition. This is a therapeutic target for Covid-19 treatments, and the creation of rapid/easy detection assays for detecting inhibition have value for the current pandemic and will likely be valuable for other present or future viruses. The paper is well presented and the authors robustly characterize their sensor.

Comment 1: Figure 3G: The inset blocks a portion of the y-axis and its labelling. Please move it to make the axis clear.

Response: We would like to thank the reviewer for the suggestion. We have now moved the inset for clarity and would like to note that the y-axis in the graph is one and the same for both the short and long Mpro sensor constructs i.e. data for both sensor constructs have been plotted in the same graph. We have edited the legend to indicate the same.

The figure and the legend have been modified in the following way:

Figure 3. Efficient cleavage of the M^{pro} sensor constructs in live cells. (A,B) Schematic showing the M^{pro} sensor constructs - short (A) and long (B) – with SARS-CoV-2 M^{pro} N-terminal autocleavage sequence. (C,D) Graph showing a representative bioluminescence spectra of the short (C) and long (D) M^{pro} sensor constructs either in control cells or in cells expressing the WT or C145A mutant M^{pro}. Data were fit to a two Gaussian model reflecting mNG fluorescence and NLuc bioluminescence peaks. Note the reduction in the mNG peak (533 nm) of both the short and the long sensors when coexpressed with the wild type M^{pro} while no reduction was observed when coexpressed with the C145A mutant M^{pro}. (E,F) Graphs showing total mNG fluorescence (measured prior to substrate addition) in cells expressing the short (E) and the long (F) sensors. (G) Graph showing BRET ratio (ratio emission at 533 nm and 467 nm) of the short (left side) and the long (right side) M^{pro} protease activity sensors in either control cells or when coexpressed with the wild type or the C145A mutant M^{pro}. **Note that data for both the short and the long sensor have been plotted on the same y-axis for the ease of comparison. Indicated p-values were obtained from unpaired, two-sided Student's t-test.** Inset: graph showing percentage change in BRET of the short (left side) and the long (right side) when co-expressed with the wild type or the C145A mutant M^{pro}. **Data shown are mean ± S.D. from three independent experiments, each performed in triplicates.** (H) Top panel: anti-His tag blot showing cleavage of the short (left side) and the long (right side) M^{pro} sensor constructs in either control cells or in cells co-expressing the wild type or the C145A mutant M^{pro}. Note the release of an approximately 30 kDa, His₆-tagged-mNG fragment in cells expressing the wild type but not in the C145A mutant M^{pro}. Bottom panel: anti-Strep-tag blot showing expression of the M^{pro} in the respectively transfected cells.

Comment 2: Figure 3: The authors state: "Data shown are mean \pm S.D. from a representative of independent experiments performed multiple times". However, all data should be shown. The methods suggest two independent replicates each containing 3 technical replicates, and the presence of only 3 data points looks like only one set of technical replicates were graphed.

Response: We would like to thank the reviewer for raising this point. We have now included data from 3 independent experiments (mean values obtained from each experiment performed in triplicates) and used them to determine mean BRET and % change in BRET (in the inset).

The figure legend has been modified in the following way:

"Data shown are mean \pm S.D. from three independent experiments, each performed in triplicates."

The figure has been modified in the following way:

Figure 3. Efficient cleavage of the M^{pro} sensor constructs in live cells. (A,B) Schematic showing the M^{pro} sensor constructs - short (A) and long (B) – with SARS-CoV-2 M^{pro} N-terminal autocleavage sequence. (C,D) Graph showing a representative bioluminescence spectra of the short (C) and long (D) M^{pro} sensor constructs either in control cells or in cells expressing the WT or C145A mutant M^{pro}. Data were fit to a two Gaussian model reflecting mNG fluorescence and NLuc bioluminescence peaks. Note the reduction in the mNG peak (533 nm) of both the short and the long sensors when coexpressed with the wild type M^{pro} while no reduction was observed when coexpressed with the C145A mutant M^{pro}. (E,F) Graphs showing total mNG fluorescence (measured prior to substrate addition) in cells expressing the short (E) and the long (F) sensors. (G) Graph showing BRET ratio (ratio emission at 533 nm and 467 nm) of the short (left side) and the long (right side) M^{pro} protease activity sensors in either control cells or when coexpressed with the wild type or the C145A mutant M^{pro}. **Note that data for both the short and the long sensor have been plotted using the same y-axis for the ease of comparison. Indicated p-values were obtained from unpaired, two-sided Student's t-test. Inset: graph showing percentage change in BRET of the short (left side) and the long (right side) when co-expressed with the wild type or the C145A mutant M^{pro}. Data shown are mean ± S.D. from three independent experiments, each performed in triplicates.** (H) Top panel: anti-His tag blot showing cleavage of the short (left side) and the long (right side) M^{pro} sensor constructs in either control cells or in cells co-expressing the wild type or the C145A mutant M^{pro}. Note the release of an approximately 30 kDa, His₆-tagged-mNG fragment in cells expressing the wild type but not in the C145A mutant M^{pro}. Bottom panel: anti-Strep-tag blot showing expression of the M^{pro} in the respectively transfected cells.

Comment 3: Figures 4-7 are likewise unclear on the number of replicates being represented and appear to be a subset of the data collected.

Response: We would like to thank the Reviewer for raising this point and apologies for the typos with the number of experiments in the figure legends.

The figure describing the M^{Pro} proteolytic activity using FlipGFP assay (Fig.5 of previous version) is now included in the Supplementary figures (Supp. Fig. S7).

We have now included data from all three independent experiments, each performed in triplicates, in the insets of Figures 4 & 5. The figures and their respective legends have been modified to include the data from all three experiments.

Figure 5. **M^{pro} inhibition monitored in live cells using the BRET-based M^{pro} sensors.** (A,B) Graphs showing GC376-mediated inhibition of M^{pro} proteolytic cleavage of the short (A) and long (B) M^{pro} sensor in live cells. Data shown are mean \pm S.D. from a representative of three independent experiments, with each experiment performed in triplicates. Insets: graphs showing percentage proteolytic activity of WT M^{pro} in live cells (mean \pm S.D.) and fitted to sigmoidal dose-response curve from three independent experiments. Bar graphs in the insets show log(IC₅₀) obtained from the three experiments. (C) Graph showing % GFP⁺ cells expressing the FlipGFP-based M^{pro} sensor and wild type M^{pro} at the indicated concentrations of GC376. Inset, graph showing log(IC₅₀) values (mean \pm S.D. of two independent experiments). Outsets, graphs showing % GFP⁺ cells expressing the C145A mutant M^{pro} (left) or in the absence of M^{pro} (right). Data shown are mean \pm S.D. of two independent experiments.

We have now included data from two independent experiments, each performed in duplicates, in the insets of Figures 6. The figures and their respective legends have been modified to include the data from both experiments.

Imaging data of the FlipGFP experiment has now been moved to the supplementary information (Supp. Fig. S7).

Supp. Fig. S7. Monitoring M^{pro} proteolytic activity using the FlipGFP-based M^{pro} sensor in live cells.

(A) Epifluorescence images of cells showing time-dependent expression of GFP, which is converted from the non-fluorescent FlipGFP upon proteolytic cleavage by M^{pro} (top panel), mCherry (middle panel) and merge (bottom panel) in cells transfected with the WT M^{pro} . (B) Graphs showing GFP and mCherry fluorescence in individual cells transfected with the WT M^{pro} at the indicated time points. (C) Epifluorescence images of cells showing time-dependent expression of GFP (top panel), mCherry (middle panel) and merge (bottom panel) in cells transfected with the C145A mutant M^{pro} . (D) Graphs showing GFP and mCherry fluorescence in individual cells transfected with the C145A mutant M^{pro} at the indicated time points. (E) Graph showing % GFP⁺ cells at the indicated time post transfection with either the wild type or the C145A mutant M^{pro} .

Comment 4: Lines 354-428: Perhaps place this in the supplementary materials. It seems to be validating the authors construct choice, but does not add much value considering the authors rigorously tested both experimentally.

Response: We would like to thank the Reviewer for the kind words on experimental validation of the sensor constructs and the suggestion. However, we would prefer to keep the MD simulation data in the main text.

Comment 5: Lines 489 to 531: This experiment is unclear to me. The methods suggest an empty “filler plasmid” was also transformed. This sounds like there will be cells transformed with the filler plasmid rather than Mpro, which isn't really a dose-response of Mpro on the reporter. How do the authors know this is a dose response of Mpro rather than just having more cells transformed with the filler plasmid instead of Mpro?

Response: We would like to thank the Reviewer for raising this point. We have now made changes in the text to indicate that we used a control, pcDNA3.1-based plasmid DNA (not expressing Mpro) as a transfection control to maintain the total amount of DNA used for transfection. We agree with the Reviewer in that it is not possible to control the amount of expression of the protease in individual cells in this way. However, we would like to indicate that the experiments were performed with a population of cells and the measurements were also performed at the population level, and the dose-response of Mpro is at the level of the cell population. We would like to note that the transfection of cells with the control plasmid does not affect the function of the sensor which is evident from the “No protease” control. Our data also show an inverse relation between the concentration of M^{pro} WT and sensor BRET ratio indicating the increase in protease activity, while the BRET ratio remains unchanged in the presence of mutant M^{pro} C145A (Fig. 4A,B).

The materials and methods section has been edited in the following way:

“For dose-response experiments, a control plasmid (a pcDNA3.1-based plasmid not expressing M^{pro}) is also co-transfected to maintain the amount of plasmid DNA in transfection constant. In case of time-course experiments with either the short or the long sensor, control cells (No M^{pro}) were transfected with the control plasmid while the wild type (WT) and the C145A mutant M^{pro} cells were transfected with the respective M^{pro} expressing plasmid DNA. The time-course experiments were carried out at 1:5 sensor-to-protease plasmid DNA ratio.”

Comment 6: Lines 33-36, 124-126,594-611, 740-741: I think the present authors did not perform enough robust experiments to make these claims of increased specificity and sensitivity. The present authors performed one experiment and showed the FlipGFP-based sensor exhibits more background than their one. Background is not that important here because the signal to noise ratio and detection limit are more important for determining specificity or sensitivity. In fact, the opposite appears to be true. The authors of the FlipGFP paper show an IC50 of 5.5 μ M with GC376 (See figure 3D), whereas the present paper found a much higher IC50 of 127 and 194 μ M when testing their sensor with GC376.

Response: We would like to thank the Reviewer for raising this issue. We have now included data from two independent experiments conducted with the FlipGFP-based Mpro sensor in the main text.

The text has been modified in the following way:

“Epifluorescence imaging of the cells post 4 h of transfection revealed the appearance of GFP fluorescence in the transfected cells, as ascertained from mCherry fluorescence, in the presence of WT M^{pro} after 24 h of transfection ($67 \pm 7\%$; mean \pm S.D., n=2) while more cells showed GFP fluorescence after 48 h of transfection ($84 \pm 2\%$; mean \pm S.D., n=2) (Supp. Fig. S7). These data indicate a delayed response of the FlipGFP sensor to M^{pro} proteolytic activity in comparison to the BRET-based sensor. Additionally, a significant number of cells were found to be GFP positive after 48 h ($11 \pm 6\%$; mean \pm S.D., n=2) of FlipGFP transfection in the presence of the C145A mutant M^{pro} (Supp. Fig. S7). This is contrast to the observations made with the BRET-based sensor in the presence of the mutant M^{pro} (Fig. 4C,D).”

The figure containing images from one of the experiments has been moved to the supplementary information.

I think the present authors did not perform enough robust experiments to make these claims of increased specificity and sensitivity. The present authors performed one experiment and showed the FlipGFP-based sensor exhibits more background than their one. Background is not that important here because the signal to noise ratio and detection limit are more important for determining specificity or sensitivity. In fact, the opposite appears to be true. The authors of the FlipGFP paper show an IC₅₀ of 5.5 μ M with GC376 (See figure 3D), whereas the present paper found a much higher IC₅₀ of 127 and 194 μ M when testing their sensor with GC376.

We agree with the Reviewer in that a high background is not as important as the signal to noise ratio. However, as shown in panel E, the FlipGFP-based sensor showed some GFP+ cells (we have included the detailed ImageJ/Fiji scrip used for the determination of GFP+ cells in the supplementary) in the presence of the C145A mutant Mpro whereas the BRET-based sensor didn't show any decrease in signal under the same condition. On the IC₅₀ value, we are not certain as to why we are observing a higher value compared to the FlipGFP (data included in panel F). One possibility that we have mentioned in the main text is that the peptide presented to BRET sensor constructs resemble the native polyprotein better than either the bare minimum FRET-based peptides or the differently structured peptide in the FlipGFP sensor. We would like to note that we could observe a significantly difference in the IC₅₀ value in the presence of molecular crowding agent in vitro (Fig. 7G, H).

Comment 7: Figure 1: The mNG and NLuc on top of the proteins is very hard to read, and the authors might want to give a quick explanation of these in the figure legend.

Response: We would like to thank the Reviewer for this suggestion.

We have now edited the figure (and the legend) with increased font size for mNG and NLuc:

Comment 8: The referencing needs to be checked in this paper. For instance, should the reference at Line 597 be 29 instead of 30?

Response: We would like to thank the Reviewer for pointing out this. Since reference no. 29 also involves the FlipGFP-based sensor, we have included both reference numbers 29 and 30 at the end of this line.

The text has been modified in the following way:

“Having established the monitoring of expression-dependent proteolytic activity of M^{pro} in live cells, we then performed similar experiments with the recently reported FlipGFP-based M^{pro} proteolytic activity reporter^{29,30} to compare their performance of the biosensors in reporting M^{pro} proteolytic activity in live cells.”

Comment 9: I like that the authors included the protein sequences in the supplementary materials. Could they please add DNA sequences as well? Expression can be dependent on the exact DNA sequence and would increase the reproducibility of their work.

Response: We would like to thank the Reviewer for this suggestion and have included the full plasmid DNA sequences of the sensors in the supplementary information document.

Reviewer #2

This manuscript reported the design of Bioluminescence Resonance Energy Transfer (BRET)-based assay for the SARS-CoV-2 main protease (Mpro). Both the shorter and longer substrates were examined, which gave similar results. The ratio of reporter plasmid to the Mpro, as well as the induction time, were optimized. Flip-GFP assay was performed in parallel. It was found that the BRET assay was more sensitive and specific than the Flip-GFP assay. The optimized BRET assay was used to test Mpro inhibitor GC-376 and found to have much weaker potency than the values reported in the literature. The BRET assay was also performed in cell culture, and addition of 25% PEG 20K increased the proteolytic activity of Mpro and decreased the potency of GC-376. It was therefore claimed that Mpro is more active in the crowded environment of an infected host cell compared to in vitro conditions, thus requiring higher drug concentration for complete inhibition. Highlights of this study including the detailed assay optimization and vigorous assay calibration using the C145A dead mutants, the direct comparison with the Flip-GFP assay, and the molecular crowding experiment. To further strength the conclusions, the authors might consider the following suggestions:

Comment 1: “This is especially relevant given that the binding of the peptide substrate has been reported to allosterically activate the SARS-CoV-1 Mpro dimer.”
Comment: reference should be given.

Response: We would like to thank the Reviewer for this suggestion. We have now included the references related to the involvement of substrate in the dimerization and activation of Mpro.

The text has been modified in the following way:

“This is especially relevant given that the binding of the peptide substrate has been reported to allosterically activate the SARS-CoV-1 M^{pro} dimer.^{92,93}”

Comment 2: Is there any internal control for the BRET assay to normalize the transfection efficiency? In the Flip-GFP assay, mCherry is the internal control.

Response: We would like to thank the Reviewer for the question and would like to mention that we have used mNeonGreen fluorescence (measured upon excitation with an external source of light, instead of NLuc bioluminescence; indicated as total mNG fluorescence; Fig. 3E,F) as a way to compare the expression of the sensors, in addition to the western blot analysis. More importantly, BRET is measured as a ratio of acceptor and donor emission and therefore, is internally controlled.

Comment 3: “Additionally, a significant number of cells were found to be GFP positive after 24 h (9 ± 1%)

and 48 h ($20 \pm 1\%$) of FlipGFP transfection in the presence of the C145A mutant Mpro (Fig. 5C,D). This is contrast to the observations made with the BRET-based sensor in the presence of the mutant Mpro (Fig. 4C,D).”

Comment: the background GFP signal from the Flip-GFP assay might be a result of cleavage by the host proteases. If this the case, the BRET assay should have similar background signal as both assays contain the same substrate. Is there any explanation why the BRET assay has less leakage signal?

Response: We would like to thank the Reviewer for raising this question and much like the Reviewer, we too are perplexed as to why there is a difference between the two sensors. One possible that we could think of is that the cleavage peptide in the FlipGFP-based sensor is in a greater degree of structural constraint due to its presence in a structured protein (GFP) than that in the BRET-based sensors reported here. One may suggest that due to this, the cleavage peptide could be cleaved non-specifically by host cell proteases. Another possible reason we can think of is that while the cleavage sequence is the same in both types of sensors, additional residues spanning cleavage sequence in the sensor may result in the formation of a cleavage site for a host protease.

The text has been modified (including the new set of data) in the following way:

“Additionally, a significant number of cells were found to be GFP positive after 48 h ($11 \pm 6\%$; mean \pm S.D., n=2) of FlipGFP transfection in the presence of the C145A mutant M^{pro} (Supp. Fig. S7).”

Comment 4: “The lower efficacy of GC376 observed here compared to previous reports perhaps indicates a cell type- or Mpro 646 expression dependent effect.”

Comment: HEK 293T cells were used in both the BRET and Flip-GFP assay, so the above statement does not hold. For direct comparison, the author should also determine the EC50 of GC-376 in the Flip-GFP assay. This is to rule out the possibility of incorrect drug concentration or the different cell type used in this study from the ones reported in the literature. Another possibility might be the drug efflux pump P-gp and GC-376 is a known substrate of P-gp (ACS Infect. Dis. 2021, 7, 3, 586–597). However, this is unlikely as 293T is not known to have high levels of P-gp.

Response: We would like to thank the Reviewer for raising this concern. We have now performed live cell Mpro inhibition assay using the FlipGFP sensor and find that the GC376 is IC50 similar to that reported previously². We have now edited Fig. 5 to include this.

Figure 5. **M^{pro} inhibition monitored in live cells using the BRET-based M^{pro} sensors.** (A,B) Graphs showing GC376-mediated inhibition of M^{pro} proteolytic cleavage of the short (A) and long (B) M^{pro} sensor in live cells. Data shown are mean \pm S.D. from a representative of three independent experiments, with each experiment performed in triplicates. Insets: graphs showing percentage proteolytic activity of WT M^{pro} in live cells (mean \pm S.D.) and fitted to sigmoidal dose-response curve from three independent experiments. Bar graphs in the insets show log(IC₅₀) obtained from the three experiments. (C) Graph showing % GFP⁺ cells expressing the FlipGFP-based M^{pro} sensor and wild type M^{pro} at the indicated concentrations of GC376. Inset, graph showing log(IC₅₀) values (mean \pm S.D. of two independent experiments). Outsets, graphs showing % GFP⁺ cells expressing the C145A mutant M^{pro} (left) or in the absence of M^{pro} (right). Data shown are mean \pm S.D. of two independent experiments.

We have edited the main text in the following way:

“These data showed that the IC₅₀ values for the short sensor is $127.4 \pm 23.33 \mu\text{M}$ and that for long sensor is $194.7 \pm 7.49 \mu\text{M}$ while the IC₅₀ value obtained using the FlipGFP sensor was $5.453 \pm 1.03 \mu\text{M}$ (Fig. S8). The lower efficacy of GC376 observed here with the BRET-based sensor compared to the FlipGFP-based sensor² (and previous reports^{2-6,27,58}) perhaps indicates that the peptide substrate presented sandwiched between the mNG and NLuc proteins serves as a better substrate than either in the FRET-based peptides or in the FlipGFP sensor construct.”

Comment 5: “Together, these data indicate that Mpro could be more active in the crowded environment of an infected host cell compared to in vitro conditions, and may require higher concentrations of pharmacological inhibitors for effective inhibitions of its catalytic activity than those determined from in vitro assays.”

Comment: To provide additional evidence for this conclusion, the authors should repeat the FRET assay with and without 25% PEG 20K. In addition, the lack of direct correlation between the results of in vitro assay and the cell-based assay might due to many factors including cell membrane permeability, drug efflux, protein binding, off-target effect, metabolism and etc.

Response: We would like to thank the Reviewer for this excellent suggestion and would like to note that such activation of Mpro has been previously shown using the FRET assay⁷. We agree with the Reviewer that molecular crowding in cells may not entirely explain the differences we observed in in vitro and live cell assays. However, the difference in the IC₅₀ values obtained using the BRET-based sensors and the FlipGFP-based sensor (Fig. S8) likely indicates that the BRET-based sensors serve as better substrates for Mpro and thus, one may need to use higher concentrations of the Mpro inhibitors, especially under the crowded environment of living cells. We have edited the main text in the following way:

“While a number of factors including cell membrane permeability, drug efflux, interaction with proteins, off-target effects and metabolism of the drug can impact efficacy of a drug, results from live cell assays with the BRET-based sensors and the FlipGFP-based sensor combined with those from in vitro assays suggest that the molecular crowding-mediated M^{pro} activation and the better presentation of the substrate peptide in the BRET-based sensor constructs could be the probable reasons for the increased IC₅₀ value of GC376 for SARS-CoV-2 M^{pro} determined using the BRET-based sensors. This then posits that higher concentrations of pharmacological inhibitors may be required for effective inhibitions of M^{pro} catalytic activity in living cells than those determined from in vitro assays.”

References:

1. Xue, X.; Yu, H.; Yang, H.; Xue, F.; Wu, Z.; Shen, W.; Li, J.; Zhou, Z.; Ding, Y.; Zhao, Q.; Zhang, X. C.; Liao, M.; Bartlam, M.; Rao, Z., Structures of two coronavirus main proteases: implications for substrate binding and antiviral drug design. *J. Virol.* **2008**, *82* (5), 2515-27.
2. Froggatt, H. M.; Heaton, B. E.; Heaton, N. S., Development of a Fluorescence-Based, High-Throughput SARS-CoV-2 3CL(pro) Reporter Assay. *J. Virol.* **2020**, *94* (22).
3. Ma, C.; Sacco, M. D.; Hurst, B.; Townsend, J. A.; Hu, Y.; Szeto, T.; Zhang, X.; Tarbet, B.; Marty, M. T.; Chen, Y.; Wang, J., Boceprevir, GC-376, and calpain inhibitors II, XII inhibit SARS-CoV-2 viral replication by targeting the viral main protease. *Cell Res* **2020**, *30* (8), 678-692.
4. Fu, L.; Ye, F.; Feng, Y.; Yu, F.; Wang, Q.; Wu, Y.; Zhao, C.; Sun, H.; Huang, B.; Niu, P.; Song, H.; Shi, Y.; Li, X.; Tan, W.; Qi, J.; Gao, G. F., Both Boceprevir and GC376 efficaciously inhibit SARS-CoV-2 by targeting its main protease. *Nat. Commun.* **2020**, *11* (1), 4417.
5. Dai, W.; Zhang, B.; Jiang, X. M.; Su, H.; Li, J.; Zhao, Y.; Xie, X.; Jin, Z.; Peng, J.; Liu, F.; Li, C.; Li, Y.; Bai, F.; Wang, H.; Cheng, X.; Cen, X.; Hu, S.; Yang, X.; Wang, J.; Liu, X.; Xiao, G.; Jiang, H.; Rao, Z.; Zhang, L. K.; Xu, Y.; Yang, H.; Liu, H., Structure-based design of antiviral drug candidates targeting the SARS-CoV-2 main protease. *Science* **2020**, *368* (6497), 1331-1335.
6. Sacco, M. D.; Ma, C.; Lagarias, P.; Gao, A.; Townsend, J. A.; Meng, X.; Dube, P.; Zhang, X.; Hu, Y.; Kitamura, N.; Hurst, B.; Tarbet, B.; Marty, M. T.; Kolocouris, A.; Xiang, Y.; Chen, Y.; Wang, J., Structure and inhibition of the SARS-CoV-2 main protease reveal strategy for developing dual inhibitors against M^{pro} and cathepsin L. *Science Advances* **2020**, *6* (50), eabe0751.
7. Okamoto, D. N.; Oliveira, L. C.; Kondo, M. Y.; Cezari, M. H.; Szeltner, Z.; Juhasz, T.; Juliano, M. A.; Polgar, L.; Juliano, L.; Gouvea, I. E., Increase of SARS-CoV 3CL peptidase activity due to macromolecular crowding effects in the milieu composition. *Biol. Chem.* **2010**, *391* (12), 1461-8.

Reviewers' comments:

Reviewer #1 (Remarks to the Author):

The authors addressed most of the comments, however I still have concerns around two points.

Response to Reviewer 1, comment 6:

The authors seem confused on the meaning of specificity and sensitivity. Below are the definitions of these terms.

- Sensitivity (true positive rate) refers to the probability of a positive test, conditioned on truly being positive.
- Specificity (true negative rate) refers to the probability of a negative test, conditioned on truly being negative.

The authors show no experiments looking at the chance of true positives or false negatives. The reduced background the authors discuss again does not affect the ability to distinguish true positives and negatives (and is therefore unrelated to specificity and sensitivity). Any assay has background and that is why researchers use negative controls.

If anything, the authors seem to show the FlipGFP assay to work better by being able to detect GC376 at a 24X lower concentration ($5.453 \pm 1.03 \mu\text{M}$ vs. $127.4 \pm 23.33 \mu\text{M}$). In the absence of proper experiments, my guess is that this would translate to better sensitivity and specificity for the FlipGFP assay.

Response to Reviewer 1, comment 5:

Thank you for clarifying the experiment. The authors did not answer how this is a dose response of Mpro rather than just having more cells transformed with the filler plasmid instead of Mpro?

The setup of the experiment means that when lower Mpro plasmid concentrations are used there will be more cells containing only the BRET sensor and not Mpro. It follows that these cells cannot undergo the BRET reaction and would be expected to create the result seen. I find the description of this as a dose-dependency of Mpro to be incorrect when all they do is create less cells with Mpro. The authors should make it clear that what they are doing is altering the proportion of cells that contain both Mpro and BRET, and not refer to this as a potency of Mpro.

Reviewer #2 (Remarks to the Author):

Comments from the previous round of review were properly addressed. I therefore recommend acceptance.

Second Revision

Reviewers' comments:

Reviewer #1 (Remarks to the Author):

The authors addressed most of the comments, however I still have concerns around two points.

Reviewer 1, comment 6:

The authors seem confused on the meaning of specificity and sensitivity. Below are the definitions of these terms.

- Sensitivity (true positive rate) refers to the probability of a positive test, conditioned on truly being positive.
- Specificity (true negative rate) refers to the probability of a negative test, conditioned on truly being negative.

The authors show no experiments looking at the chance of true positives or false negatives. The reduced background the authors discuss again does not affect the ability to distinguish true positives and negatives (and is therefore unrelated to specificity and sensitivity). Any assay has background and that is why researchers use negative controls.

If anything, the authors seem to show the FlipGFP assay to work better by being able to detect GC376 at a 24X lower concentration ($5.453 \pm 1.03 \mu\text{M}$ vs. $127.4 \pm 23.33 \mu\text{M}$). In the absence of proper experiments, my guess is that this would translate to better sensitivity and specificity for the FlipGFP assay.

Response: We would like to thank the Reviewer for raising these points and elaborating on the definition of assay sensitivity and specificity.

First, we have performed the live cell GC376 dose response experiments with the BRET-based sensor using a modified protocol wherein we treat the cells with various concentrations of GC376 right from the time of transfection of the sensor and Mpro plasmids, instead of allowing them to be expressed for 8 hours before initiating inhibitor treatment of the cells. Additionally, we measured BRET after 16 hours of transfection and inhibition, instead of 24 hours of transfection/inhibition. The IC50 values determined using this modified protocol ($9.80 \pm 3.84 \mu\text{M}$ and $17.86 \pm 2.14 \mu\text{M}$ for the short and the longer sensor, respectively) agrees with that obtained using the FlipGFP-based Mpro sensor ($5.453 \pm 1.03 \mu\text{M}$) and those reported previously. The higher IC50 values obtained in experiments performed using the previous protocol is likely due to cleavage of the sensors during their maturation in the initial stages of the after transfections and resolves the issue with IC50 values highlighted by the Reviewer.

Second, we generally agree with the Reviewer on the issue of assay sensitivity and specificity. However, we would like to point out that we have performed experiments with the catalytically dead mutant (C145A) Mpro and have shown that there is no reduction in the BRET ratio in the presence of the mutant Mpro while a large, robust decrease in the BRET ratio was observed in the presence of the wild type Mpro (Figure 3G; for both the short and the long sensor), likely indicating the specificity of the assay.

We have edited the main text at several places and updated both Figure 5 and Supporting Figure 4.

The text has been modified in the following ways:

Abstract:

Here, we report a pair of genetically encoded, Bioluminescence Resonance Energy Transfer (BRET)-based sensors for detecting SARS-CoV-2 main protease (M^{pro}), which is critical for its replication and a target of anti-SARS-CoV-2 agents, proteolytic activity in living host cells as well as in vitro assays. The sensors were generated by sandwiching M^{pro} N-terminal autocleavage sites, either AVLQSGFR (short) or KTS AVLQSGFRKME (long), in between the mNeonGreen and nanoLuc proteins. Co-expression of the sensors with M^{pro} in live cells resulted in their cleavage while mutation of the critical C145 residue (C145A) in M^{pro} completely abrogated their cleavage. Additionally, the sensors recapitulated the inhibition of M^{pro} by the well-characterized pharmacological agent GC376. Further, in vitro assays with the BRET-based M^{pro} sensors revealed a molecular crowding-mediated increase in the rate of M^{pro} activity and a decrease in the inhibitory potential of GC376. The sensors developed here will find direct utility in studies related to drug discovery targeting the SARS-CoV-2 M^{pro} and functional genomics application to determine the effect of sequence variation in M^{pro} .

Last paragraph in the introduction section:

The sensor constructs showed robust cleavage activity in live cells when coexpressed with the wild type M^{pro} but not in the presence of the catalytically dead C145A mutant M^{pro1-3} , and with a faster kinetics and higher specificity compared to the recently reported FlipGFP-based M^{pro} sensor.

Live cell Mpro proteolytic cleavage inhibitor assay – in the Methods section

HEK 293T cells were co-transfected with either pmNG- M^{pro} -Nter-auto-NLuc or pmNG- M^{pro} -Nter-auto-L-NLuc plasmid along with either pLVX-EF1alpha-SARS-CoV-2-nsp5-2xStrep-IRES-Puro (M^{pro} WT) (Addgene plasmid # 141370) or pLVX-EF1alpha-SARS-CoV-2-nsp5-C145A-2xStrep-IRES-Puro (M^{pro} C145A) (Addgene plasmid # 141371) plasmid in 96-well white flat bottom plates at a ratio of 1:5 of sensor-to-protease plasmid DNA ratio. Transfected cells were concomitantly treated with a range of GC376 (GC376 Sodium; AOBIOUS- AOB36447; stock solution prepared in 50% DMSO at a concentration of 10 mM) concentrations. After 16 h of incubation with the inhibitor, BRET measurements were performed by the addition of furimazine (Promega, Wisconsin, USA) at a dilution of 1:200.

Monitoring pharmacological inhibition of Mpro proteolytic activity in live cells – in the Results section

These data showed that the IC_{50} values for the short sensor is $9.80 \pm 3.84 \mu M$ and that for long sensor is $17.86 \pm 2.14 \mu M$ in a general agreement with the IC_{50} value obtained using the FlipGFP sensor⁴ ($5.453 \pm 1.03 \mu M$; Fig. 5C) and those reported previously^{4-8,27,58}.

In vitro assays reveal molecular crowding-mediated increase in Mpro proteolytic activity and a decrease in inhibitor efficacy – in the Results section

These IC_{50} values are in agreement with those obtained from live cell assays and suggest that molecular crowding prevalent in living cells might activate M^{pro} and impact the inhibitory potential of M^{pro} inhibitors developed to treat COVID-19

Conclusion

~~Deleted:~~ The BRET-based sensors developed here appears to possess better sensitivity and specificity compared to the FlipGFP-based M^{Pro} .

Updated Figure 5:

Updated Supporting Figure 4:

A mNG-M^{pro}-Nter-auto-NLuc
(Short M^{pro} Sensor)

B mNG-M^{pro}-Nter-auto-L-NLuc
(Long M^{pro} Sensor)

Response to Reviewer 1, comment 5:

Thank you for clarifying the experiment. The authors did not answer how this is a dose response of Mpro rather than just having more cells transformed with the filler plasmid instead of Mpro?

The setup of the experiment means that when lower Mpro plasmid concentrations are used there will be more cells containing only the BRET sensor and not Mpro. It follows that these cells cannot undergo the BRET reaction and would be expected to create the result seen. I find the description of this as a dose-dependency of Mpro to be incorrect when all they do is create less cells with Mpro. The authors should make it clear that what they are doing is altering the proportion of cells that contain both Mpro and BRET, and not refer to this as a potency of Mpro.

Response: We would like to thank the Reviewer for highlighting the issue with the usage of the term “dose-response” in experiments where we have used different concentration of Mpro plasmid to monitor sensor cleavage and agree completely with the Reviewer in that by using increasing concentrations of the Mpro plasmid DNA (replacing the control, non-Mpro expressing plasmid DNA), we are increasing the number of cells transfected with Mpro plasmid, and thus, increasing number of cells expressing the protease (which is apparent from the decrease in BRET ratio with increasing DNA concentration). We note that our experiments were performed at the population level, instead of single cell level, and therefore, we chose to use the term “Mpro DNA dose-response”. While it is true that the use of a higher concentration/amount of Mpro plasmid DNA will result in the increase in the number of cells transfected with the Mpro plasmid (and thus, expressing Mpro), it is also possible that a fraction of cells within the population of cells are transfected with multiple copies of the Mpro plasmid DNA. Either way, this is resulting in an increased cleavage of the BRET-based sensors suggesting that the sensor cleavage is likely due to Mpro expression (this is in addition to our data with the C145A mutant Mpro).

We have now made changes in this section in the following way:

M^{pro} DNA concentration-dependent cleavage of the sensor in live cells

Having established that the BRET ratio could be used to detect M^{pro} proteolytic activity of the sensor constructs, we aimed to determine the M^{pro} DNA concentration-dependent cleavage of the sensor constructs in live cells. For this, we cotransfected cells with the 25 ng/well sensor constructs and a range of M^{pro} plasmid concentrations (0, 0.0125, 0.125, 1.25, 12.5 and 125 ng/well) to gradually increase the number of cells expressing M^{pro} and monitored bioluminescence spectra in adherent cells after 48 h. This revealed a M^{pro} plasmid DNA concentration-dependent shift in the bioluminescence spectra (Supp. Fig. S4) and BRET ratios (Fig. 4A,B) of both the short and the long sensor in the presence of the wild type M^{pro} but not in the presence of the C145A mutant M^{pro}. Discernable decreases in the BRET ratio could be observed at a minimum amount of 1.25 ng/well of M^{pro} plasmid DNA and a maximum decrease in the BRET ratio of ~100% at the highest concentration of 125 ng/well for both sensor constructs (Fig. 4A,B; insets). The analysis has also showed that the EC₅₀ values are 1.77 ± 1.07 ng/well and 1.85 ± 1.01 ng/well for the short and the long sensors, respectively. These data demonstrate the functional potency of M^{pro} expressed in these cells.

Reviewer #2 (Remarks to the Author):

Comments from the previous round of review were properly addressed. I therefore recommend acceptance.

Response: We would like to thank Reviewer for all the comments and suggestions resulting in the improvement of the manuscript.

References:

1. Gordon, D. E.; Jang, G. M.; Bouhaddou, M.; Xu, J.; Obernier, K.; White, K. M.; O'Meara, M. J.; Rezelj, V. V.; Guo, J. Z.; Swaney, D. L.; Tummino, T. A.; Huettenhain, R.; Kaake, R. M.; Richards, A. L.; Tutuncuoglu, B.; Foussard, H.; Batra, J.; Haas, K.; Modak, M.; Kim, M.; Haas, P.; Polacco, B. J.; Braberg, H.; Fabius, J. M.; Eckhardt, M.; Soucheray, M.; Bennett, M. J.; Cakir, M.; McGregor, M. J.; Li, Q.; Meyer, B.; Roesch, F.; Vallet, T.; Mac Kain, A.; Miorin, L.; Moreno, E.; Naing, Z. Z. C.; Zhou, Y.; Peng, S.; Shi, Y.; Zhang, Z.; Shen, W.; Kirby, I. T.; Melnyk, J. E.; Chorba, J. S.; Lou, K.; Dai, S. A.; Barrio-Hernandez, I.; Memon, D.; Hernandez-Armenta, C.; Lyu, J.; Mathy, C. J. P.; Perica, T.; Pilla, K. B.; Ganesan, S. J.; Saltzberg, D. J.; Rakesh, R.; Liu, X.; Rosenthal, S. B.; Calviello, L.; Venkataramanan, S.; Liboy-Lugo, J.; Lin, Y.; Huang, X. P.; Liu, Y.; Wankowicz, S. A.; Bohn, M.; Safari, M.; Ugur, F. S.; Koh, C.; Savar, N. S.; Tran, Q. D.; Shengjuler, D.; Fletcher, S. J.; O'Neal, M. C.; Cai, Y.; Chang, J. C. J.; Broadhurst, D. J.; Klippsten, S.; Sharp, P. P.; Wenzell, N. A.; Kuzuoglu, D.; Wang, H. Y.; Trenker, R.; Young, J. M.; Caverro, D. A.; Hiatt, J.; Roth, T. L.; Rathore, U.; Subramanian, A.; Noack, J.; Hubert, M.; Stroud, R. M.; Frankel, A. D.; Rosenberg, O. S.; Verba, K. A.; Agard, D. A.; Ott, M.; Emerman, M.; Jura, N.; von Zastrow, M.; Verdin, E.; Ashworth, A.; Schwartz, O.; d'Enfert, C.; Mukherjee, S.; Jacobson, M.; Malik, H. S.; Fujimori, D. G.; Ideker, T.; Craik, C. S.; Floor, S. N.; Fraser, J. S.; Gross, J. D.; Sali, A.; Roth, B. L.; Ruggero, D.; Taunton, J.; Kortemme, T.; Beltrao, P.; Vignuzzi, M.; Garcia-Sastre, A.; Shokat, K. M.; Shoichet, B. K.; Krogan, N. J., A SARS-CoV-2 protein interaction map reveals targets for drug repurposing. *Nature* **2020**.
2. Huang, C.; Wei, P.; Fan, K.; Liu, Y.; Lai, L., 3C-like proteinase from SARS coronavirus catalyzes substrate hydrolysis by a general base mechanism. *Biochemistry* **2004**, *43* (15), 4568-74.
3. Shan, Y. F.; Li, S. F.; Xu, G. J., A novel auto-cleavage assay for studying mutational effects on the active site of severe acute respiratory syndrome coronavirus 3C-like protease. *Biochem Biophys Res Commun* **2004**, *324* (2), 579-83.
4. Froggatt, H. M.; Heaton, B. E.; Heaton, N. S., Development of a Fluorescence-Based, High-Throughput SARS-CoV-2 3CL(pro) Reporter Assay. *J. Virol.* **2020**, *94* (22).
5. Ma, C.; Sacco, M. D.; Hurst, B.; Townsend, J. A.; Hu, Y.; Szeto, T.; Zhang, X.; Tarbet, B.; Marty, M. T.; Chen, Y.; Wang, J., Boceprevir, GC-376, and calpain inhibitors II, XII inhibit SARS-CoV-2 viral replication by targeting the viral main protease. *Cell Res* **2020**, *30* (8), 678-692.
6. Fu, L.; Ye, F.; Feng, Y.; Yu, F.; Wang, Q.; Wu, Y.; Zhao, C.; Sun, H.; Huang, B.; Niu, P.; Song, H.; Shi, Y.; Li, X.; Tan, W.; Qi, J.; Gao, G. F., Both Boceprevir and GC376 efficaciously inhibit SARS-CoV-2 by targeting its main protease. *Nat. Commun.* **2020**, *11* (1), 4417.
7. Dai, W.; Zhang, B.; Jiang, X. M.; Su, H.; Li, J.; Zhao, Y.; Xie, X.; Jin, Z.; Peng, J.; Liu, F.; Li, C.; Li, Y.; Bai, F.; Wang, H.; Cheng, X.; Cen, X.; Hu, S.; Yang, X.; Wang, J.; Liu, X.; Xiao, G.; Jiang, H.; Rao, Z.; Zhang, L. K.; Xu, Y.; Yang, H.; Liu, H., Structure-based design of antiviral drug candidates targeting the SARS-CoV-2 main protease. *Science* **2020**, *368* (6497), 1331-1335.
8. Sacco, M. D.; Ma, C.; Lagarias, P.; Gao, A.; Townsend, J. A.; Meng, X.; Dube, P.; Zhang, X.; Hu, Y.; Kitamura, N.; Hurst, B.; Tarbet, B.; Marty, M. T.; Kolocouris, A.; Xiang, Y.; Chen, Y.; Wang, J., Structure and inhibition of the SARS-CoV-2 main protease reveal

strategy for developing dual inhibitors against M^{pro} and cathepsin L. *Science Advances* **2020**, *6* (50), eabe0751.

REVIEWERS' COMMENTS:

Reviewer #1 (Remarks to the Author):

The authors addressed all of my concerns. I recommend this paper for acceptance.